

# Wind Tunnel Load Measurements of a Leading-Edge Inflatable Kite Rigid Scale Model

Jelle Agatho Wilhelm Poland[1], Johannes Marinus van Spronsen[1], Mac Gaunaa[2], and Roland Schmehl[1]

[1]Faculty of Aerospace Engineering, Delft University of Technology, Kluyverweg 1, 2629 HS, Delft, the Netherlands
[2]Department of Wind and Energy Systems, Technical University of Denmark, Frederiksborgvej 399, 4000 Roskilde, Denmark

**Correspondence:** Jelle Agatho Wilhelm Poland (j.a.w.poland@tudelft.nl)

**Abstract.** Leading-edge inflatable (LEI) kites are morphing aerodynamic surfaces that are actuated by the bridle line system. Their design as tensile membrane structures has several implications for the aerodynamic performance. Because of the pronounced C-shape of the wings, a considerable part of the aerodynamic forces is redirected sideways and used for steering. The inflated tubular frame introduces flow recirculation zones on the pressure side of the wing. In this paper, we present wind tunnel
measurements of a 1:6.5 rigid scale model of the 25 m$^2$ TU Delft V3 LEI kite developed specifically for airborne wind energy (AWE) harvesting. Because the real kite deforms during flight, the scale model was manufactured to match the well-defined design geometry. Aerodynamic forces and moments were recorded in an open jet wind tunnel over large ranges of angles of attack and sideslip, for five different inflow speeds. The wind tunnel measurements were performed with and without zigzag tape along the model's leading edge to investigate the possible boundary layer tripping effect of the stitching seam connecting the canopy to the inflated tube. To quantify the quality of the acquired data, the autocorrelation-consistent confidence inter-
vals, coefficient of variation, and measurement repeatability were reported, and the effects of sensor drift and flow-induced vibrations of the test setup at the highest Reynolds number were assessed. A representative subset of the measurements was compared to Reynolds-averaged Navier-Stokes (RANS) flow simulations from literature, as well as new simulations conducted with an existing Vortex-Step Method (VSM). In conclusion, the measured aerodynamic characteristics validate both RANS and
VSM simulations under nominal kite operating conditions, with both models yielding similar trends and values within a 5 to 10% range.

## 1 Introduction

Airborne wind energy (AWE) systems use tethered flying devices to capture wind energy. The innovative technology promises to save up to 90% of the material mass of conventional wind turbines (Van Hagen et al., 2023; Coutinho, 2024), resulting in
a lower environmental footprint and potentially lower costs while providing access to previously untapped wind resources at higher altitudes (Bechtle et al., 2019; Kleidon, 2021). A prominent concept, that is also highly mobile, uses the pulling force of a soft kite maneuvered in cross-wind patterns to drive a ground-based drum-generator module (Vermillion et al., 2021; Fagiano et al., 2022). Figure 1(a) illustrates the components of such an AWE system equipped with a leading-edge inflatable (LEI) kite with suspended kite control unit (KCU). To provide a continuous power output, the kite is operated in pumping cycles. During





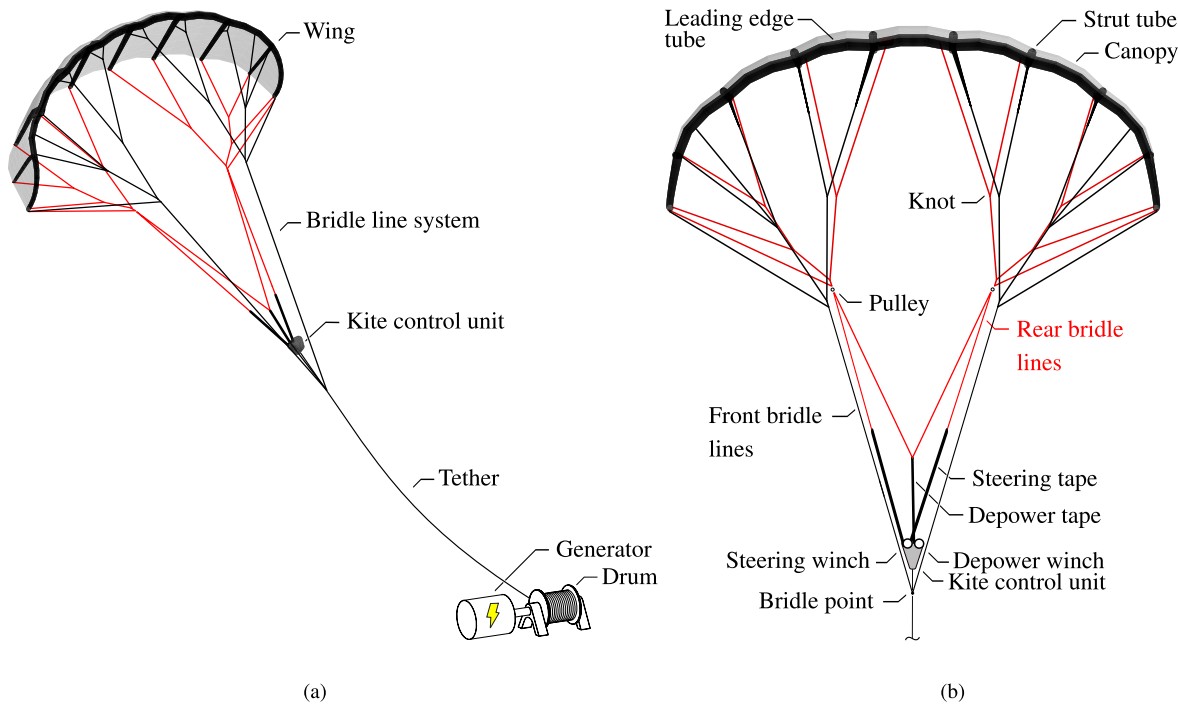

(a)                                                                          (b)

**Figure 1.** Ground-generating AWE system based on the TU Delft V3 kite, initially designed for a 20 kW technology demonstrator that was first used in 2012: (**a**) System overview, with tether and ground station depicted only schematically; (**b**) Components of the kite, consisting of wing, bridle line system and kite control. Adapted from Poland and Schmehl (2023).

the reel-out phase, the kite is guided in cross-wind flight patterns with its wing pitched to a high angle of attack. Once the tether reaches its maximum length, the cross-wind patterns are terminated, the wing is pitched to a low angle of attack and the tether is retracted, using some of the previously generated and buffered energy. The cyclic operation results in a net energy gain because the aerodynamic force during the reel-out phase is substantially larger than the force in the reel-in phase, which is also shorter than the reel-out phase.

Figure 1(a) further details the components and actuation layout of the kite. The KCU pitches and morphs the wing by adjusting the lengths of the rear bridle lines via the steering and depower tapes. Besides this actuation-induced deformation, the tensile membrane structure is also subject to strong aero-structural coupling (Oehler and Schmehl, 2019). The tubular frame of the wing consists of an inflatable leading edge tube and several connected inflatable strut tubes. This frame provides structural stability for handling on the ground and for launching and landing, and, once the kite is in flight, it transmits the

aerodynamic forces from the canopy to the bridle line system (Poland and Schmehl, 2023).

       An optimal kite design can be regarded as an effective compromise between pulling force and controllability, acknowledging that both competing properties are tightly coupled. For instance, increasing the aspect ratio will generally increase the pulling



force but decrease the agility of the kite. Similarly, making the wing flatter will increase its pulling force but decrease its steerability.

The aerodynamic properties of a kite have a major influence on the amount of wind energy that can be harvested. Accordingly, these properties play an important role in kite design, performance estimations, failure load prediction, and stability analysis for ensuring reliable and robust operation. A common approach for aerodynamic system identification is based on flight experiments. One option that provides reasonable control over the inflow conditions is towing a small kite along a straight track to measure lift, drag, and dynamic response (Dadd et al., 2010; Python, 2017; Hummel et al., 2019; Rushdi et al.,
2020; Elfert et al., 2024). A second option, applicable to larger industrial-scale kites, involves directly using sensor data from an operating AWE system to determine forces, position, and inflow conditions (Schmidt et al., 2017; Van der Vlugt et al., 2019; Oehler and Schmehl, 2019; Roullier, 2020; Schelbergen and Schmehl, 2024; Cayon et al., 2025). However, in-flight experiments are expensive, risky, and offer limited control over inflow conditions.

A less expensive, safer, and more scalable alternative is numerical simulation, which, due to actuation-induced morphing and
strong aero-structural coupling, generally requires iterative resolution of both aerodynamic and structural mechanics (Breukels, 2011; Leloup et al., 2013; Bosch et al., 2014; Duport, 2018; Van Til et al., 2018). However, simulations necessitate validation, which is best achieved through wind tunnel testing that allows precise control of inflow conditions. Although wind tunnel experiments for LEI kites have not been reported in the public literature, related soft-wing structures have been described, including sail airfoil sections (Den Boer, 1980), paragliders (Nicolaides, 1971; Matos et al., 1998; Babinsky, 1999), ram-air
wings (Wachter, 2008; Rementeria Zalduegui and Garry, 2019), and inflatable wings (Cocke, 1958; Smith et al., 2007; Okda et al., 2020; Desai et al., 2024).

One significant challenge for wind tunnel studies of industrial kites is that these membrane structures ranging from 50 to 500 $m^2$ do not fit inside standard tunnels and thus require scaling. Aeroelastic effects complicate scaling because maintaining the correct proportion of structural to aerodynamic loads is non-trivial, as highlighted by Oehler et al. (2018). Additionally, de-
veloping such models encounters manufacturing and material limitations; for instance, adjusting beam bending stiffness would necessitate impractically high inflation pressures. Lastly, comparing experimental data to aero-structural coupled simulations lacks specificity, making it unclear whether discrepancies arise from errors in modeling aerodynamics, structural dynamics, coupling mechanisms, or other factors.

Wind tunnel experiments using rigid kite models eliminate the aeroelastic scaling issues and provide aerodynamic data with
a high degree of certainty on the inflow. Belloc (2015) presented wind tunnel measurements of a 1:8 scale paraglider model in which the anhedral angle follows an elliptical shape when viewed from the front, and it incorporates a spar made of a wood–carbon composite sandwich. During the tests, inflow velocities reached 40 $ms^{-1}$, corresponding to Reynolds numbers of 9.2 $\times 10^5$. The experiments covered angles of attack ranging from −5 to 22° and sideslip angles from −15 to 15°.

Omitting deformation isolates the aerodynamic problem and provides the necessary specificity to validate simulations. The
literature reports LEI kite aerodynamic simulations ranging from low-fidelity potential flow methods to high-fidelity computational fluid dynamic (CFD) methods. The potential flow methods are often a form of Prandtl (1918) lifting-line theory, and to increase accuracy, most models include the addition of nonlinear section lift-curve slopes, i.e., airfoil polars (Leloup et al.,



2013; De Solminihac et al., 2018; Cayon et al., 2023). The airfoil polar aerodynamic simulations should incorporate viscosity and vorticity to accurately represent the generally present separation zone aft of the inflatable tube, e.g., using Reynolds-Average Navier Stokes (RANS) CFD (Breukels, 2011; Folkersma et al., 2019; Watchorn, 2023). RANS CFD simulations have also been conducted in three dimensions for the TU Delft V2 kite (Deaves, 2015) and for the V3 kite with and without struts (Viré et al., 2020, 2022).

The present paper is based on the graduation project of Van Spronsen (2024), presenting a novel wind tunnel experiment of an LEI kite to acquire validation data for numerical tools. The aerodynamic characteristics of a rigid scale model of the V3 kite are obtained over an extensive range of inflow conditions, with a high degree of certainty on the match between simulated and measured geometry and inflow conditions. Thorough analysis of potential sources of uncertainty reinforces the reliability of the measured aerodynamic loads. In addition, the effects of forced boundary layer transition, Reynolds number variation, and sideslip are examined in detail. Measured aerodynamic forces and moments are compared with numerical simulations to assess the consistency between experimental and computational results.

The remainder of this paper is organized as follows. Section 2 describes the experimental methodology. Section 3 presents the results of our wind tunnel tests, focusing on analyzing the uncertainties and the effect of Reynolds number. A discussion on the agreement with numerical predictions follows in Sect. 4, and the conclusions are presented in Sect. 5 along with recommendations for future work.

## 2 Experimental methodology

This section discusses the specifics of the wind tunnel and the scale model. This is followed by a description of the experimental setup, the measurement matrix, zig-zag tape measurements, and the data processing method, including the required wind tunnel corrections.

### 2.1 Open Jet Facility

The wind tunnel experiments were conducted in the Open Jet Facility (OJF) at the Faculty of Aerospace Engineering of Delft University of Technology from 1 to 10 April 2024. The facility is a closed-loop wind tunnel, featuring an octagonal jet exhaust nozzle with maximum dimensions of $2.85 \times 2.85$ m, and a contraction ratio of 3:1, as illustrated in Fig 2. The jet discharges into a test section room with dimensions $13$ m in width and $8$ m in height. The wind tunnel is equipped with a $500$ kW electric motor driving a large fan, which generates a controlled streamwise velocity of up to $35$ ms$^{-1}$ in the test section. Corner vanes and wire meshes guide the flow to ensure uniform flow conditions, resulting in a turbulence intensity of 0.5% in the test section (Lignarolo et al., 2014).

### 2.2 Rigid scale model

As the original TU Delft LEI V3 kite is 8.3 m wide and the width of the OJF exhaust nozzle is only 2.85 m, a scale model had to be used. With the main purpose of the measurement campaign being the acquisition of validation data for numerical





**Figure 2.** CAD drawing of the experimental setup, showing the origin $O$ in the load balance representing the point at which the load measurements are made. The x-axis runs along the longitudinal direction of the wind tunnel, pointing downstream parallel to the wind. The y-axis is oriented laterally, pointing to the left when facing downstream. The z-axis is vertical, pointing upwards. The rotary table, load balance, support structure, and kite are all placed on the blue table, which was adjusted in lateral position and height to center the model in the nozzle exit.





tools, the scale model was manufactured to match the wing geometry used in earlier CFD simulations (Viré et al., 2022). This
geometry differs from the original CAD geometry in several aspects: it does not include the bridle line system, the trailing
edge connecting upper and lower canopy surfaces is rounded, and an edge fillet is applied at all canopy-tube connections. The
model geometry was verified using a laser tracker with a spatial resolution of 0.5 μm (FARO, 2024). Figure 3 compares the
manufactured physical model with the rendering of the geometry and the overlaid laser-tracked outline of the physical model.

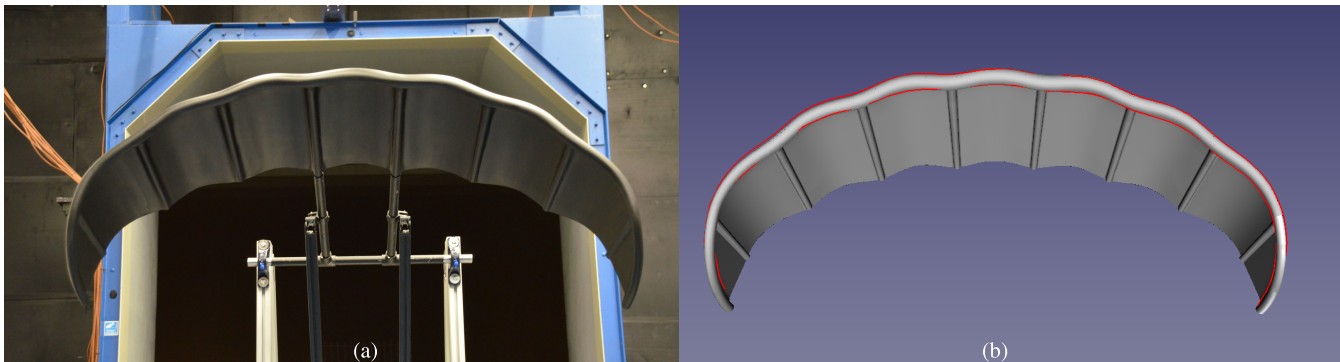

**Figure 3.** Rigid scale model of the TU Delft LEI V3 kite: (a) Photo of the model, rotated by 180° with its back facing the blue octagonal
OJF exhaust nozzle; (b) Rendering of the model, from a similar perspective, with overlaid the laser-tracked outline in red.

**Table 1.** Properties of the rigid scale model, including values for the physical scale model and the scaled design geometry. The physical
model properties were measured using a laser tracker, while the scaled design geometry values correspond to the scaled design geometry of
the kite. The relative error between the physical model and the scaled design geometry is also provided.

| Property | Symbol | Unit | Physical Scale Model | Scaled Design Geometry | Relative Error (%) |
|---|---|---|---|---|---|
| Midspan chord | $c_{ref}$ | m | 0.395 | 0.396 | 0.25 |
| Height | $h$ | m | 0.462 | 0.462 | 0.00 |
| Width | $w$ | m | 1.278 | 1.277 | 0.08 |
| Mass | $m$ | kg | 7.965 | - | - |
| Flat surface area | $S$ | m$^2$ | - | 0.59 | - |
| Planform area | $A$ | m$^2$ | - | 0.46 | - |
| Projected frontal area at $\alpha = 24°$ | $A_f$ | m$^2$ | - | 0.2 | - |

Considering manufacturing costs, handling limitations, Reynolds number scaling, and wind tunnel blockages, we decided
on a 1:6.5 scaling of the wind tunnel model, leading to the dimensions listed in Table 1. The anhedral swept wing with a bow-
shaped leading edge and double-curved canopy was manufactured by Curveworks B.V. using carbon fiber reinforced plastic
layed-up in a 3D-milled mold from structural foam. The canopy is 3 mm thick, except for the two central panels, which are 4
mm as they need to sustain a higher load. The outer layers provided the most structural support and were made of carbon fiber.





The 1 or 2 mm inner layers were made of a glass fiber-reinforced polymer. Structural foam was used inside the chordwise

struts, except for the two inner struts, which incorporate two parallel steel rods. These rods slide into the two aluminum sleeve

tubes of the support frame, as illustrated in Fig. 3(a).

## 2.3   Measurement equipment

The support frame is a truss structure assembled from custom-cut aluminum profiles. The angle of attack $\alpha$ quantifies the

inclination of the mid-span chord line with respect to the inflow, and can be adjusted as illustrated in Fig. 4. The angle was

measured with an accuracy of $0.1°$ by placing two digital inclinometers on the aluminum sleeve tubes. The measured value

is converted to the angle of attack $\alpha$ by subtracting the offset angle $6.3°$ between the chord line and the parallel steel rods of

the model. The support structure was placed aft of the kite to minimize flow interference and mounted onto a 6-component

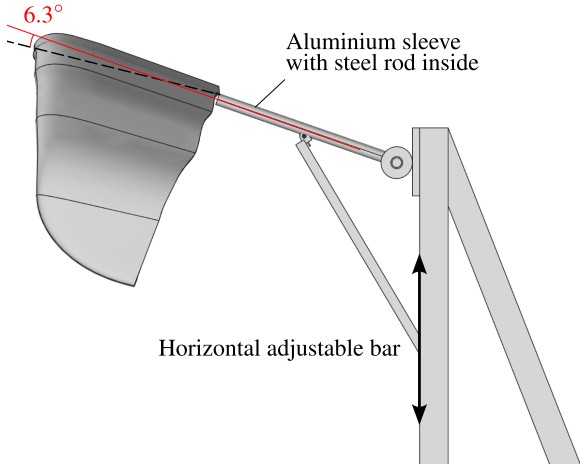

**Figure 4.** Manual setting of the scale model's angle of attack with respect to the inflow by adjusting the vertical position of the strut
attachment to the support structure. The center of gravity of the scale model is indicated by point CG.

load balance, as illustrated in Fig. 2. The balance operates at 2000 Hz and is equipped with six load cells able to measure

the longitudinal, inflow-aligned ($F_x$), transverse ($F_y$), and vertical ($F_z$) forces, and the roll ($M_x$), pitch ($M_y$) and yaw ($M_z$)

moments. The entire assembly was mounted on a rotary table, allowing a remote adjustment of the side slip angle $\beta$ with an

angular resolution of $0.01°$. The side slip angle is defined with respect to the origin $O$ and positive in the positive yaw direction.

## 2.4   Measurement matrix

The experiments were conducted for most combinations of $\alpha$, $\beta$, and $U_\infty$ values displayed in Table 2. Due to time constraints,

not all $\beta$ values were tested for each $\alpha$ value. The Reynolds number,

$$\text{Re} = \frac{U_\infty c_{\text{ref}}}{\nu},\tag{1}$$





**Table 2.** Parametric combinations investigated with wind tunnel measurements.

| Parameter | Range |
|---|---|
| Angle of attack $\alpha$ (°) | $-11.6, -6.1, -2.0, -1.3, 3.1, 5.4,$ $7.4, 9.4, 11.5, 12.5, 13.4, 14.5, 16.2,$ $18.3, 20.2, 23.0, 24.5$ |
| Inflow speed $U_\infty$ (ms$^{-1}$) | $5, 10, 15, 20, 25$ |
| Reynolds number $\mathrm{Re}/10^5$ (−) | $1.3, 2.5, 3.8, 5.0, 6.1$ |
| Side slip $\beta$ (°) | $-20, -14, -12, -10, -8, -6, -4, -2,$ $0, 2, 4, 6, 8, 10, 14, 20$ |

is used to characterize the flow regime, recalculating the kinematic viscosity $\nu$ for each value of $U_\infty$ using Sutherland's law (Poling et al., 2001). The characteristic aerodynamic time (Flay and Jackson, 1992), defined as the ratio between $c_{\mathrm{ref}}$ and $U_\infty$, represents the time for a fluid element to travel along the reference chord length of the kite. A measuring period of 10 s, resulting in 125 to 625 fluid parcel passings depending on the used $U_\infty$, was thus deemed a statistically sufficient sampling period. Measurements without the kite were made over the full range of parameters to quantify the aerodynamic loads on the support structure only. The interference effects between the support structure and the kite are assumed to be negligible. To ensure consistency, measurements taken with $\alpha = 5.7°$, at $U_\infty = 20 \,\mathrm{ms}^{-1}$ and $\beta = -20, 0,$ and $20°$ were repeated three times. Furthermore, the sensor drift of the force balance during the campaign was analyzed through six measurements done over a 30 s time interval each morning and evening with $U_\infty = 0 \,\mathrm{ms}^{-1}$ for three consecutive days.

## 2.5 Laminar-turbulent flow transition

Using 2D CFD simulations, Folkersma et al. (2019) showed that incorporating a boundary layer transition model significantly affects the aerodynamic predictions for $\mathrm{Re} < 200 \times 10^5$. This motivated the use of natural transition modeling in subsequent 3D CFD simulations of the V3 kite (Viré et al., 2020, 2022). In practice, transition may be influenced by the zigzag-patterned stitching seam connecting the canopy to the tube along the span, as shown in Fig. 5(a). Whether this seam height would be sufficient to induce transition remained uncertain, however. To address this, additional measurements with zigzag tape, as shown in Fig. 5(b), were conducted to assess the effect of forced transition.

The critical roughness Reynolds number $\mathrm{Re}_{\mathrm{k,crit}}$ is commonly used to quantify the threshold at which a surface roughness element induces boundary layer transition. The numerical estimation of this number is nontrivial, as it depends on local pressure gradients, freestream disturbances, geometry, and roughness characteristics (Ye, 2017). In practice, trip heights are often estimated through empirical correlations (Langel et al., 2014; Gahraz et al., 2018). Braslow and Knox (1958) reported typical values of $\mathrm{Re}_{\mathrm{k,crit}}$ ranging between 300 and 600. For zigzag or wavy-patterned 2D roughness, Balakumar (2021) adopted a value of 300, while others found 200 to be sufficient (van Rooij and Timmer, 2003; Elsinga and Westerweel, 2012). Given a value of



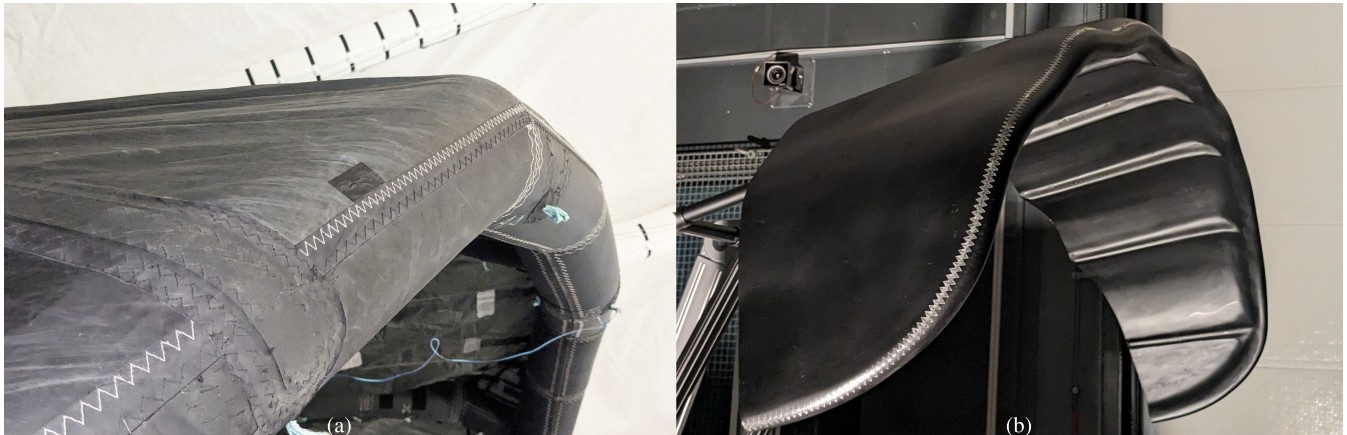

**Figure 5.** (a) Kitepower V3.25B kite with seams along the leading edge; (b) Scale model with zigzag tape applied to the leading edge. Although slightly different designs, the V3.25B and TU Delft V3 kites are practically identical with respect to the flow over the wing's suction side.

$\text{Re}_{\text{k,crit}}$, the corresponding roughness height $k$ can be computed using the relation (Braslow and Knox, 1958)

$$k = \frac{\text{Re}_{\text{k,crit}}\nu}{U_{\text{k}}}, \tag{2}$$

where $U_{\text{k}}$ is the local velocity at the roughness height, which may be approximated by $U_\infty$ (Driest and McCauley, 1960; Tani, 1969). The resulting functional dependency of $k$ on the Reynolds number defined by Eq. (1) is shown in Fig. 6 for two different values of $\text{Re}_{\text{k,crit}}$. The diagram also includes the selected tape height of $0.2$ mm to trigger transition from approximately

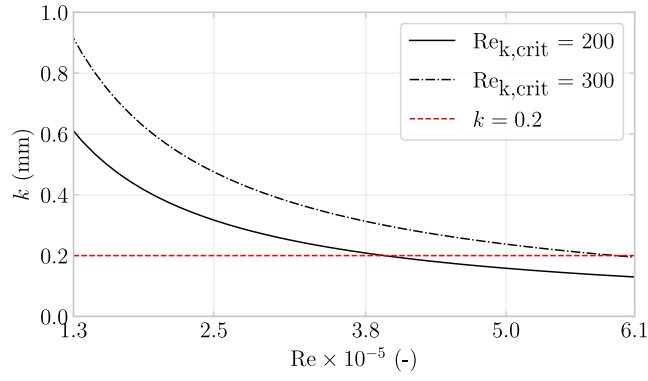

**Figure 6.** Required minimal trip height versus Re for different values of $\text{Re}_{\text{k,crit}}$.

$\text{Re} \geq 3.9 \times 10^5$ according to the estimate $\text{Re}_{\text{k,crit}} = 200$. The tape, produced by Glasfaser Flugzeug-Service GmbH with a $60°$ tooth angle, was applied at 5% chord, following the approach in Soltani et al. (2011); Gahraz et al. (2018); Dollinger et al.
(2019); De Tavernier (2021).





## 2.6  Data post-processing

The measured load data were converted to the non-dimensional aerodynamic coefficients as follows:

1. subtract zero-wind measurements,

2. non-dimensionalize the load data,


3. translate the coordinate system, from the load balance origin $O$ to the center of gravity of the scale model

4. correct for sideslip,

5. subtract non-dimensionalized support structure loads,

6. apply wind tunnel corrections.

(1) First, the zero-wind measurements taken before every $\alpha$ change were subtracted to eliminate background noise from the
signals, including the structure's weight and sensor drift.

(2) In the next step, the measurements were non-dimensionalized using the air density $\rho$, which varied from 1.14 to 1.19
$\mathrm{kgm}^{-3}$, the inflow speed $U_\infty$, the projected area $A$ and the reference chord $c_{\mathrm{ref}}$ of the scale model, listed in Table 1. The forces
$F_\mathrm{i}$ and moments $M_{\mathrm{i,b}}$ are non-dimensionalized using

$$C_\mathrm{i} = \frac{2F_\mathrm{i}}{\rho U_\infty^2 A}, \qquad\qquad i = 1, 2, 3,$$

$$C_{\mathrm{M,i,b}} = \frac{2M_{\mathrm{i,b}}}{\rho U_\infty^2 A c_{\mathrm{ref}}}, \qquad\qquad i = 1, 2, 3.$$

(3) To represent the moment coefficients in the wing reference frame, they had to be translated from the load-balance
measurement center to the center of gravity CG of the scale model. With $\alpha = 0°$, the CG is located at $-0.172$ m in $x$- and
$-0.229$ m in $z$-direction with respect to the mid-span trailing-edge point, see Fig. 4. The rolling moment coefficient $C_{\mathrm{M,x,b}}$ is
translated using,

$$C_{\mathrm{M,x}} = C_{\mathrm{M,x,b}} - C_{\mathrm{F,y}} z_{\mathrm{cg}}. \tag{3}$$

The pitching- and yawing-moment coefficients, $C_{\mathrm{M,y}}$ and $C_{\mathrm{M,z}}$, respectively, are determined as

$$C_{\mathrm{M,y}} = -C_{\mathrm{M,y,b}} + C_{\mathrm{F,z}} x_{\mathrm{cg}} - C_{\mathrm{F,x}} z_{\mathrm{cg}}, \tag{4}$$
$$C_{\mathrm{M,z}} = -C_{\mathrm{F,z,b}} - C_{\mathrm{F,y}} x_{\mathrm{cg}}. \tag{5}$$

In these expressions, $x_{\mathrm{cg}}$, $y_{\mathrm{cg}}$ and $x_{\mathrm{cg}}$ are the coordinates of the scale model's center of gravity, with respect to $O$.
(4) Because the force balance was mounted on top of the rotary table, and $y$ is defined perpendicular to the incoming flow,
the force and measured moment coefficients had to be corrected for the sideslip. The force and moment coefficient vectors are





transformed, at each sideslip angle $\beta_i$, through matrix multiplication with the rotation matrix $\mathbf{R}$:

$$
\mathbf{R} = \begin{bmatrix} \cos\beta & \sin\beta & 0 \\ -\sin\beta & \cos\beta & 0 \\ 0 & 0 & 1 \end{bmatrix} .
\tag{6}
$$

(5) To isolate the aerodynamic forces of the kite, measurements were made with only the support structure. These measurements
were performed at the minimum, mean, and maximum $\alpha$ values. Missing data points were determined by interpolation, which
was carried out by fitting two linear segments from the minimum to the mean and from the mean to the maximum, respectively.

The aerodynamic loads on the support structure only were measured and processed through steps (1) to (4) such that the
resulting aerodynamic coefficients could then be subtracted from the coefficients of the kite including the support structure.
It was critical to non-dimensionalize before subtracting these measurements, as atmospheric conditions could not be assumed
constant throughout the experiment. Specifically, during the experiment, the temperature varied between 20 and 32°C.

(6) The last step entailed applying the wind tunnel corrections that arise from blockage, streamline curvature, and downwash
or upwash in both $y$- and $z$-directions. For a detailed analysis of these effects, the reader is referred to App. A. The conclusions
are that with a blockage factor of 3%, the corrections due to blockage are negligible, which aligns with the recommendations of
Wickern (2014) to keep the blockage factor below 5% and of Barlow et al. (1999) to stay below 7.5%. Following Barlow et al.,
the corrections due to streamline curvature and downwash were calculated and found to be non-negligible, shown in Table A1,
and hence applied.

## 3 Results

This section first addresses measurement uncertainties, followed by the effect of forced boundary layer transition. Subsequently,
the aerodynamic force and moment coefficients are presented as functions of the angle of attack and sideslip angle, with the
Reynolds number as parameter.

### 3.1 Uncertainty analysis

This section quantifies the main sources of measurement uncertainty to ensure data reliability and repeatability, including sensor
drift, support-to-kite load proportion, vibration analysis, coefficient of variation, and measurement repeatability. Although
a load balance sensor drift was detected, it was concluded not to affect the results, as detailed in App. B. Analyzing the
proportions of support-structure loads to kite loads as signal-to-noise ratio, one finds high certainty for lift and lower for $C_{\mathrm{M,y}}$,
$C_{\mathrm{M,z}}$, as detailed in App. C.

For some measurements at $U_\infty = 25\,\mathrm{ms}^{-1}$ and high values of $\alpha$ and $\beta$, the wind tunnel model started to vibrate considerably.
To avoid physical damage, these specific measurements were not completed, which is why some data points are missing at
$\mathrm{Re} = 6.1 \times 10^5$. A vibration analysis revealed structural resonance at 4–5 Hz, close to the resonance frequency of the supporting
blue table (see Fig. 2) as reported in LeBlanc and Ferreira (2018). This frequency band was not filtered to avoid introducing
processing artifacts. See App. D for details, e.g., time series and power spectral density analyses.



The coefficient of variation, denoted as CV, offers a dimensionless metric for comparing variability across different datasets by normalizing the standard deviation relative to the mean (Pearson, 1896),

$$\mathrm{CV_i} = \frac{\overline{\sigma_i}}{\mu_i}, \qquad\qquad i = 1, 2, 3, 4, 5, 6$$

Table 3 lists $\mathrm{CV}_i$ for each Re, except for $\mathrm{Re} = 6.1 \times 10^5$, which is excluded due to incomplete data. The means $\mu_i$ were computed over the full range of $\alpha$ and $\beta$ for $C_{\mathrm{V,L}}$, $C_{\mathrm{V,D}}$, and $C_{\mathrm{V,M,y}}$. For $C_{\mathrm{V,S}}$, $C_{\mathrm{V,M,x}}$, and $C_{\mathrm{V,M,z}}$, only positive values of $\beta$ were considered to avoid including near-zero loads at $\beta = 0°$, which could lead to inflated values of $\mathrm{CV}_i$ and skew the statistical averages.

The decline in $\mathrm{CV}_i$ values from $\mathrm{Re} = 1.3$ to $2.5 \times 10^5$ reflects a reduction in measurement uncertainty, likely due to an im-
proved signal-to-noise ratio at higher wind speeds. Force measurements generally exhibit lower relative uncertainty compared to moment measurements, which is attributed to their inherently lower signal-to-noise ratios. The cases at $\mathrm{Re} = 3.8$ and $5 \times 10^5$ show the smallest values of $\mathrm{CV}_i$, indicating the highest measurement precision.

**Table 3.** Coefficient of variation $\mathrm{CV}_i$ of the data for varying Re.

| $\mathrm{Re} \times 10^5$ (−) | 1.3 | 2.5 | 3.8 | 5 |
|---|---|---|---|---|
| $\mathrm{CV_L}$ | 1.11 | 0.35 | 0.17 | 0.15 |
| $\mathrm{CV_D}$ | 0.84 | 0.54 | 0.53 | 0.58 |
| $\mathrm{CV_S}$ | 1.27 | 0.94 | 0.89 | 0.90 |
| $\mathrm{CV_{M,x}}$ | 5.67 | 2.28 | 2.18 | 2.31 |
| $\mathrm{CV_{M,y}}$ | 33.40 | 8.43 | 4.54 | 5.33 |
| $\mathrm{CV_{M,z}}$ | 2.90 | 2.54 | 2.90 | 2.24 |

At $\mathrm{Re} = 5 \times 10^5$, $\alpha = 5.7°$ and $\beta = -20, 0$ and $20°$ measurements were made three times to check the repeatability. For each of these measurements, the standard deviation within these repeated measurements $\sigma_{\mathrm{rm}}$ is shown in Table 4. The authors
conclude that the measurement repeatability is overall high by comparing the orders of magnitude of the averaged standard deviation, $1 \times 10^{-1}$, and the repeatability standard deviation, $1 \times 10^{-4}$. Smaller uncertainties show for $\beta = 0°$, affecting the lift coefficient $C_{\mathrm{L}}$ and the pitching moment $C_{\mathrm{M,y}}$ the most.

## 3.2  Effect of forced boundary layer transition

The measured aerodynamic force coefficients with and without zigzag tape are plotted in Fig. 7 for $\beta = 0°$ and $\alpha = 9.4°$,
excluding the $\mathrm{Re} = 6.1 \times 10^5$ case due to missing data. In addition to the mean values, a confidence interval (CI) is plotted, indicating with 99% certainty that the mean lies within the given range. As detailed in Sec. 2.4, the load balance records data over a 10 s time interval, thereby capturing between 125 and 625 fluid parcels passing through. The resulting samples are regarded as temporally correlated; one supporting argument is that each fluid element traverses the measurement region over 16 to 80 ms, while data are sampled at much finer 0.5 ms intervals. To accurately estimate the sample measurement uncertainty



**Table 4.** Standard deviations of the repeatability measurements $\sigma_{\mathrm{rm}}$ for three $\beta$ values taken with $\alpha = 5.7°$.

| | $\sigma_{\mathrm{rm}} \times 10^{-4}$ | | |
| | $\beta = -20°$ | $\beta = 0°$ | $\beta = 20°$ |
| --- | --- | --- | --- |
| $C_{\mathrm{L}}$ | 2.793 | 0.699 | 2.562 |
| $C_{\mathrm{D}}$ | 0.076 | 0.085 | 0.014 |
| $C_{\mathrm{S}}$ | 0.085 | 0.030 | 0.300 |
| $C_{\mathrm{M,x}}$ | 1.903 | 1.030 | 2.585 |
| $C_{\mathrm{M,y}}$ | 7.034 | 1.899 | 6.254 |
| $C_{\mathrm{M,z}}$ | 0.222 | 0.120 | 0.766 |

of this correlated time series, the heteroskedasticity and autocorrelation-consistent (HAC) estimator by Newey and West (1987) is employed. The method requires an estimate of the time lag. A time lag of 11 samples was found from taking the integer value of $N_{\mathrm{samples}}^{\frac{1}{4}}$ (Greene, 2019).

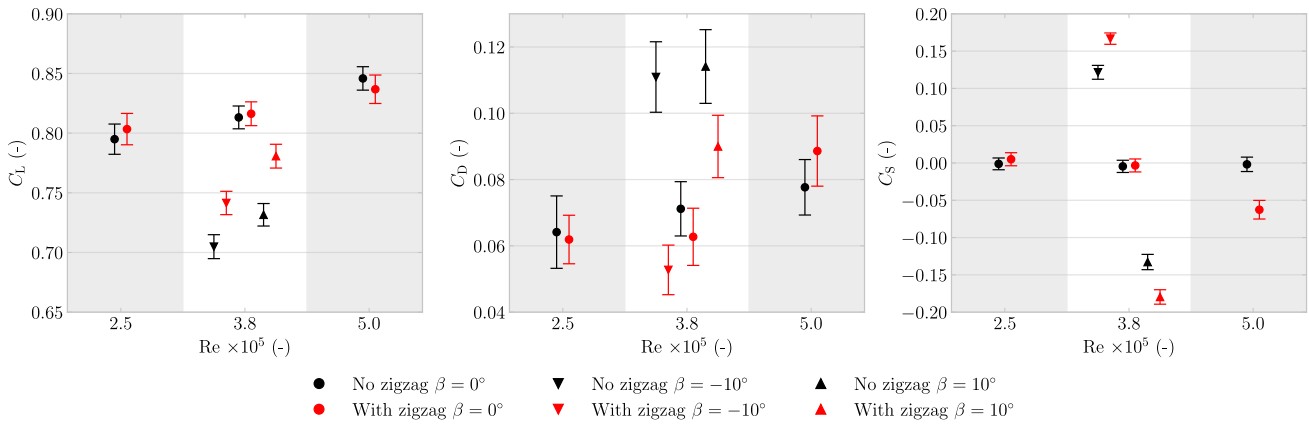

**Figure 7.** Aerodynamic force coefficients plotted with standard deviation, with and without zigzag tape, at Re $= 1.3$, $2.5$, $3.8$ and $5 \times 10^5$, at an averaged corrected $\alpha = 9.4°$.

From Fig. 6, it was concluded that a zigzag tape height of 0.2 mm would not be sufficient to force the flow to transition at Re $= 2.5 \times 10^5$, that it might be marginally sufficient at $3.8 \times 10^5$, and is sufficient at $5 \times 10^5$. At $\beta = 0°$, higher lift and

lower drag are observed for Re $= 2.5$ and $3.8 \times 10^5$, and the opposite trend is found for Re $= 5 \times 10^5$, including a 12% increase in drag. This result at Re $= 5 \times 10^5$ is consistent with findings in the literature, where the introduction of zigzag tape led to decreased lift and increased drag (Gahraz et al., 2018; Zhang et al., 2017a, b; Dollinger et al., 2019).

Definitive conclusions cannot be drawn for the non-zero sideslip cases, where data is limited to Re $= 3.8 \times 10^5$ at $\alpha = 9.4°$ for $\beta = \pm 10°$. Nonetheless, based on the measured increase in lift and side force, along with an observed 50% reduction in





drag, the authors hypothesize that, in the sideslip configuration, the zigzag tape may locally promote a laminar-to-turbulent transition that delays flow separation.

Without zigzag tape and in sideslip, the measured $C_S$ value is near zero, independent of Re. However, with zigzag tape, a negative $C_S$ is observed at Re $= 5 \times 10^5$. The difference, also visible in $C_L$ and $C_D$, suggests that the zigzag tape introduces a setup asymmetry, possibly due to imperfect tape application.

### 255  3.3   Reynolds number effects

Figure 8 shows the measured force and moment coefficients as functions of $\alpha$ for the different values of Re. With increasing Re, the measurements show a converging trend and decreasing variation, consistent with the decreasing uncertainty observed in Table 3. Furthermore, the aerodynamic performance of the wing improves with Re, i.e., higher $C_L$ and lower $C_D$, a trend that can be attributed to decreasing boundary layer thickness (Folkersma et al., 2019).

The Re $= 1.3 \times 10^5$ case exhibits the largest variations due to the less favorable signal-to-noise ratio, i.e., relatively smaller load magnitudes compared to the measurement precision and support-structure loads. The $C_L$–$\alpha$ plot suggests that, compared to higher Re, there may already be stall development at lower angles. This aligns with aerodynamic theory predicting earlier separation in laminar flows due to lower sensitivity to adverse pressure gradients (Anderson, 2016). Developing stall would also explain the increased fluctuations at higher $\alpha$, particularly in $C_S$ and the moment coefficients.

The non-zero values of $C_S$ indicate an asymmetry in the setup. This is further reflected in the moment coefficients $C_{M,x}$ and $C_{M,z}$, which both vanish under perfectly symmetric conditions. The results for Re $= 3.8 \times 10^5$ and $5 \times 10^5$ agree well, particularly the lift and drag coefficients, which both follow anticipated trends with increasing $\alpha$. The pitching moment coefficient $C_{M,y}$ also shows a consistent increase as $\alpha$ rises.

When examining the influence of $\beta$, as shown in Fig. 9 for $\alpha = 7.4°$, the largest deviations are observed at Re $= 1.3 \times 10^5$, 270   followed by Re $= 2.5 \times 10^5$. As Re increases, the results tend to converge, and the differences between the polars become less pronounced. For a perfectly symmetric setup, the coefficients $C_L$, $C_D$, and $C_{M,y}$ would be symmetric about $\beta = 0°$, while $C_S$, $C_{M,x}$, and $C_{M,z}$ would be antisymmetric. However, because of slight asymmetries in the actual setup, small deviations occur. Most notably the non-zero values of $C_S$, $C_{M,x}$, and $C_{M,z}$ at $\beta = 0°$, as well as minor asymmetries in $C_L$, $C_D$, and $C_{M,y}$ about the vertical axis.

As also evident in Fig. 8, the $C_L$ plot reveals an overall trend of improving aerodynamic performance with increasing Re. Notably, around $\beta = 8°$, the $C_S$ curve exhibits both positive and negative peaks, suggesting a non-linear relationship with $\beta$. At this same angle, a local maximum is observed in $C_{M,x}$ for the Re $= 5 \times 10^5$ case. Similar off-trend behavior near $\beta = \pm 8°$ also appears in $\beta$ sweeps at other values of $\alpha$. The potential underlying causes of this phenomenon are examined further in Sect. 4.

Among the tested cases with complete measurement sets, Re $= 5 \times 10^5$ represents the highest Reynolds number and is 280   therefore the closest to actual in-flight operational conditions, which range between Re $= 23 \times 10^5$ and $45 \times 10^5$ (Cayon et al., 2025). Consequently, this case is used as the basis for comparison with the numerical simulations. Additional arguments for choosing this specific measurement run are the low measurement uncertainty, as indicated by the $\text{CV}_i$ values in Table 3, its



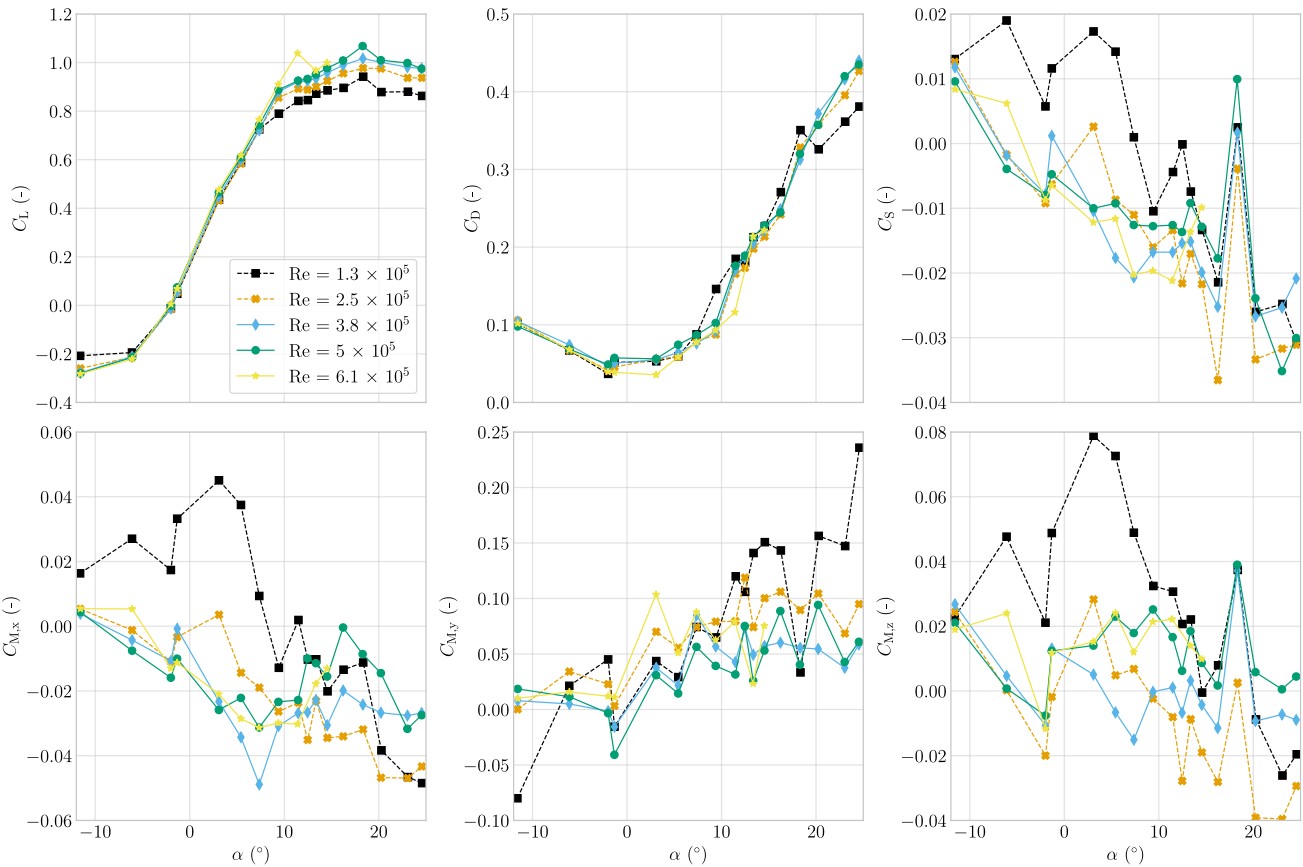

**Figure 8.** Aerodynamic force and moment coefficients plotted against $\alpha$ for Re varying from $1.3 \times 10^5$ to $6.1 \times 10^5$ at $\beta = 0°$.

high repeatability demonstrated in Table 4, and the high degree of symmetry and antisymmetry in the positive and negative $\beta$ measurements.

## 4 Discussion

Since the primary objective of the wind tunnel campaign was to generate validation data for numerical models, the measured aerodynamic characteristics were compared to characteristics obtained from several different aerodynamic computational studies of the V3 kite. One suitable data source is the Reynolds-averaged Navier-Stokes (RANS) CFD analysis by Viré et al. (2022), which is also the origin of the surface geometry employed in the present study. The closest corresponding simulation case in terms of Reynolds number is at Re $= 10 \times 10^5$, for which force data are available from both an $\alpha$ sweep at $\beta = 0°$ and a $\beta$ sweep at $\alpha = 13°$. The reported $C_S$ values differ from those reported in Viré et al. (2022) as we are using the platform area $A$, see Table 1, of the wing for the non-dimensionalization of the side force, as opposed to the projected side area that was used. For

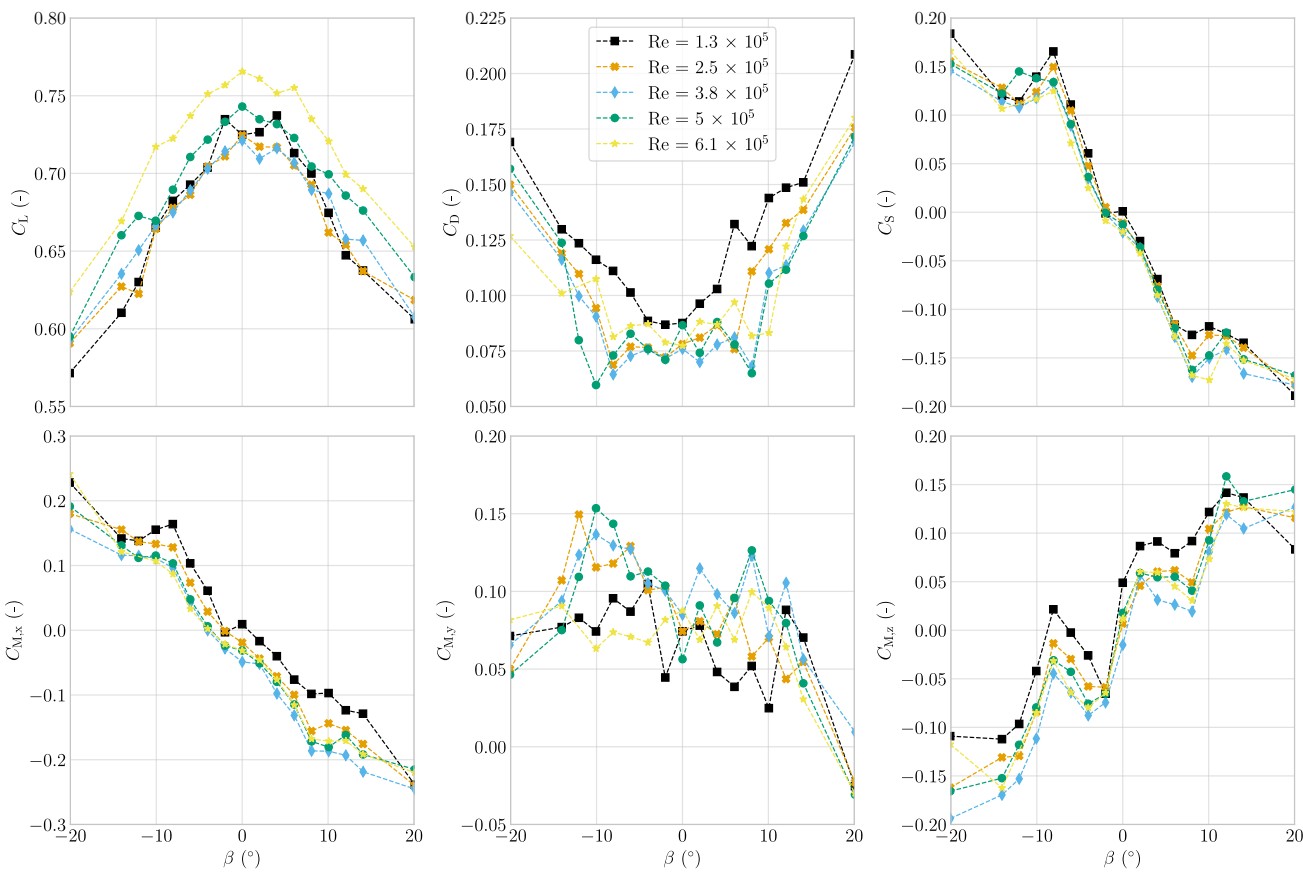

**Figure 9.** Aerodynamic force and moment coefficients plotted against $\beta$ for Re varying from $1.3 \times 10^5$ to $6.1 \times 10^5$, at $\alpha = 7.4°$.

this reason, we applied the area ratio of 3.7 as a correction factor to the values of $C_S$ reported in Viré et al. (2022). Furthermore, since RANS CFD is generally considered unsuitable for accurate modeling of unsteady separated flows (Speziale, 1998), the

post-stall residuals were examined to assess the validity of the solution. Compared to the pre-stall cases, the post-stall results exhibited larger residuals, with values ranging from $1.0 \times 10^{-5}$ to $2.2 \times 10^{-5}$, rather than remaining below $1.0 \times 10^{-6}$ as observed in pre-stall conditions. Nevertheless, these data points were retained in the analysis due to their relevance to the overall aerodynamic behavior. A second data source is the RANS CFD analysis by Viré et al. (2020) of the same wing, but without struts. This study provides force data over an $\alpha$ sweep at Re = $5 \times 10^5$. As shown by Viré et al. (2022), the struts have only a

negligible impact on the integral force coefficients of the 3D wing. For both CFD datasets from Viré et al. (2020, 2022), it was determined that the geometry file contained a $1.02°$ offset in the angle of attack, defined as the angle between the mid-span chord line and the apparent wind vector. Therefore, the numerical data presented here have been corrected by shifting the angle of attack values by this offset. The third computational dataset was generated in the present study using a Vortex-Step Method (VSM), which is a lifting-line type of method. The VSM code, originally developed by Cayon et al. (2023), was adapted for the





present comparisons; for details, see Poland et al. (2025). For each simulation, the angle of attack was incremented in steps of $1°$, and a convergence analysis confirmed that discretizing the wing into 200 spanwise panels is sufficient. The VSM relies on 2D airfoil polars as input. In previous studies, these polars were constructed using aerodynamic load correlations derived from a large set of CFD simulations (Breukels, 2011). In the present work, however, more accurate polars are employed, obtained from dedicated 2D RANS CFD simulations.

To ensure that our numerical tools accurately represent real-world flight conditions, it is essential to characterize the range of inflow angles encountered during kite operation. The experimental data indicate that the angle of attack $\alpha$ of the 3D wing averages around $1°$ during the reel-in phase and approximately $8°$ during the reel-out phase (Cayon et al., 2025). Additionally, observed sideslip angles $\beta$ typically range between $-10$ to $10°$ (Oehler et al., 2018).

## 4.1 Force comparison

In Fig. 10, the force coefficients $C_L$, $C_D$ and $L/D$ are plotted against $\alpha$, for the VSM, CFD and wind tunnel (WT) data, along with 99% CI bands, evaluated using the autocorrelation-consistent method by Newey and West (1987). The CI band is rather narrow for $C_L$ compared to the mean values, indicating high certainty. For $C_D$, the band is slightly wider, aligning with the difference in the CV listed in Table 3. The numerical data match the measured lift coefficient trend well from $\alpha = -11$ to around $11°$. Above $\alpha = 11°$, the numerical VSM predicts lower lift whereas the RANS CFD predicts higher lift. The differences in numerical predictions are considered to arise, in part, from discrepancies in turbulence modeling. The VSM employs fully turbulent two-dimensional RANS CFD as input, whereas the three-dimensional RANS CFD simulations incorporate a transition model.

The numerical and measured drag coefficients start deviating by more than a factor of two above around $\alpha = 10°$, where the lift slope also changes. The CFD predictions with and without struts agree well and show the expected change in drag slope when entering the stall regime. This effect is not reproduced by the VSM to the same extent, attributed to inherent limitations of lifting-line-based methods in this regime (Phillips and Snyder, 2000), e.g., its inviscid nature. For angles of attack above $5°$, the VSM consistently underestimates the drag coefficients compared to the measurements.

From $\alpha = 1$ to $10°$, the measured lift-to-drag ratio plateaus at the maximum value range between $L/D = 8$ and 9, sharply dropping outside this $\alpha$ range. All numerical models predict a higher maximum $L/D$: the CFD simulations reach a value of 10.5 at $\alpha = 9°$, while the VSM predicts an even higher maximum of 14 at $\alpha = 10°$.

In Fig. 11, the force coefficients $C_L$, $C_D$ and $C_S$ are plotted against $\beta$ for a low and high $\alpha$. CFD data was only available at a different angle of attack, $\alpha = 13°$, but is included in the top row of the plot to enable trend comparison. Furthermore, the WT data is plotted for both positive and negative $\beta$ ranges to illustrate the effect of the asymmetric measurement setup, e.g., due to geometry, surface condition, or inflow.

The measurements confirm and closely follow the trends predicted by the numerical simulations. With increasing side slip angle, $C_L$ decreases, while $C_D$ and the absolute value of $C_S$ both increase. The measured data at negative $\beta$ form an exception, showing an off-trend lift, drag, and side force behavior above around $\beta = 8°$. This off-trend behavior appears across multiple

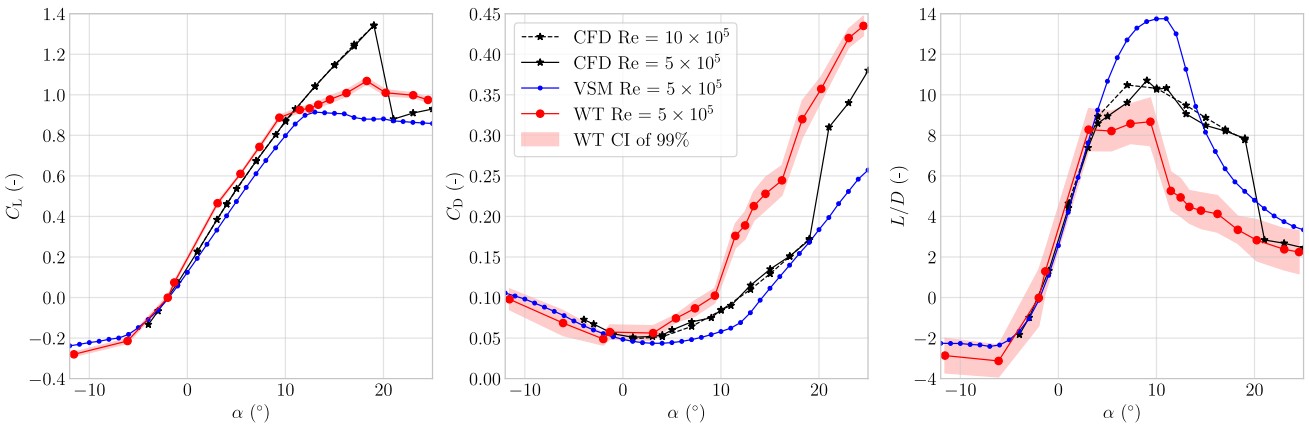

**Figure 10.** Measured lift and drag coefficients, and their ratio, together with coefficients computed with VSM and RANS CFD (Viré et al., 2020, 2022), plotted against $\alpha$ at $\beta = 0°$.

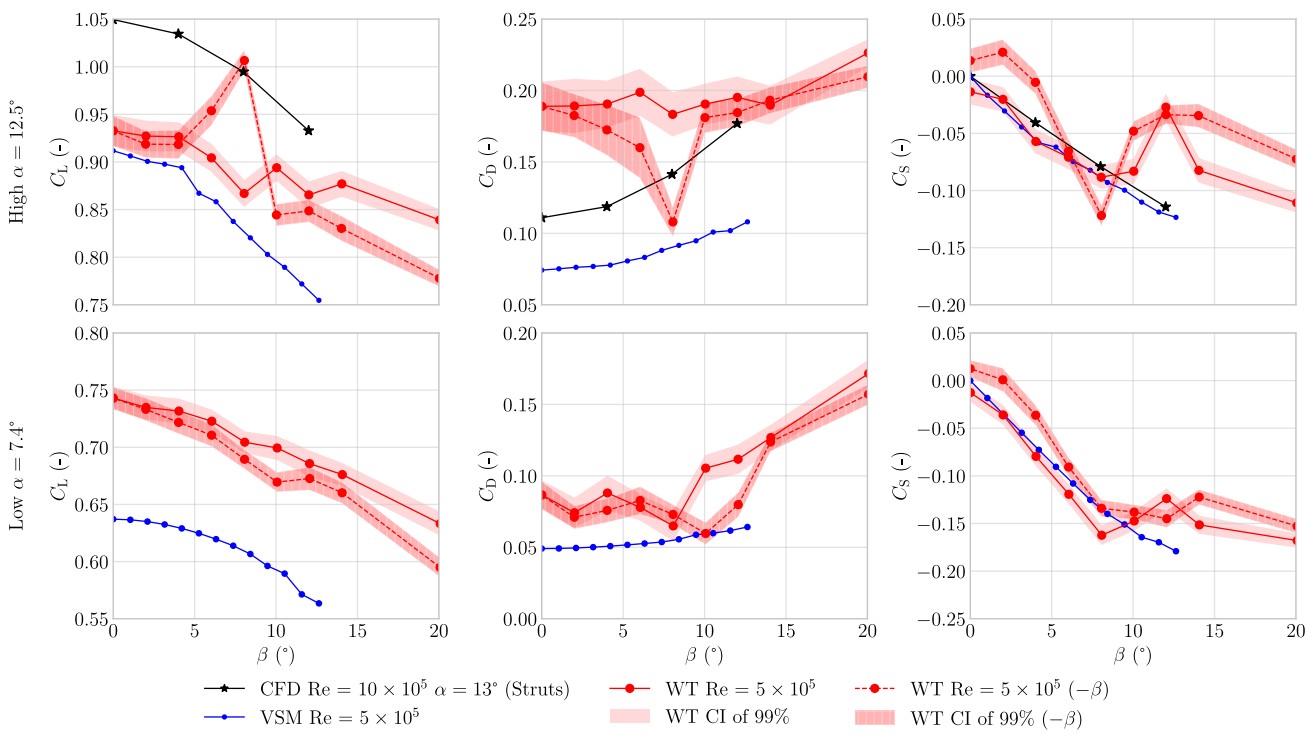

**Figure 11.** Measured lift, drag and side force coefficients, together with coefficients computed with VSM and RANS CFD Viré et al. (2022), plotted against $\beta$.





Re values, as shown in Fig. 9, and is smaller for the lower $\alpha$ case. Since it is not observed in the CFD or VSM predictions, a possible reason could be flow separation caused by imperfections in the measurement setup and the wind tunnel model.

### 340  4.2  Moment comparison

The moment coefficients $C_{M,x}$, $C_{M,y}$ and $C_{M,z}$ are plotted over an $\alpha$ sweep in Fig. 12. Compared to the force measurements, the confidence intervals are wider due to higher measurement uncertainty, the same conclusion as drawn from analyzing CV shown in Table 3. No CFD data is available; therefore, only VSM data is used. The numerical data predicts no roll or yaw moment, where the measurements do show, on average, a negative roll moment coefficient $C_{M,x}$ and a positive yaw moment coefficient

$C_{M,z}$, indicating asymmetries in the setup. The experimental pitch moment coefficients $C_{M,y}$ fluctuate significantly, yet on average exhibit a positive slope. The numerical predictions differ in magnitude but exhibit a similar positive and increasing moment trend up to the stall point.

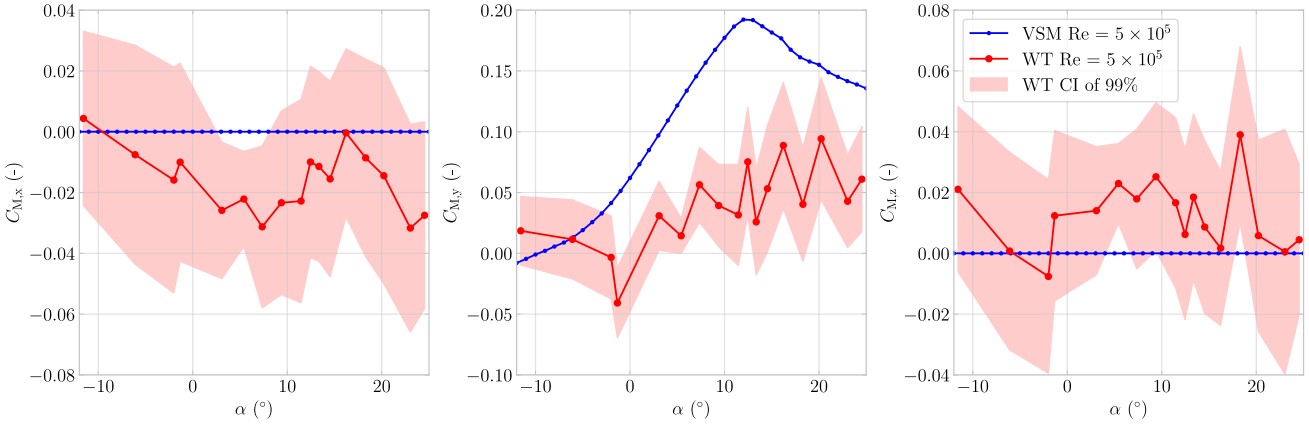

**Figure 12.** Measured and computed moment coefficients as functions of $\alpha$ at $\beta = 0°$.

In Fig. 13, the moment coefficients $C_{M,x}$, $C_{M,y}$ and $C_{M,z}$ are plotted over an $\beta$ sweep. Similar to the forces, the measured moments differ between positive and negative $\beta$ ranges. The numerical and experimental data match well for the roll moment

coefficient $C_{M,x}$. Less agreement in trend shows for the pitch moment coefficient $C_{M,y}$, where the measurements indicate an increasing moment up to $\beta = 10°$ and above this threshold a decreasing moment. The VSM, on the other hand, predicts a higher value that changes less with increasing $\beta$. For the yaw moment coefficient $C_{M,z}$, the measurements and numerical predictions show opposite trends, it is unclear why this is the case. Some possible factors contributing to this are the observed setup asymmetry, a misprediction in the aerodynamic force produced at the tips, high uncertainty as shown in Table 3, and low

signal-to-noise ratio shown in Fig. C1 in App. C.





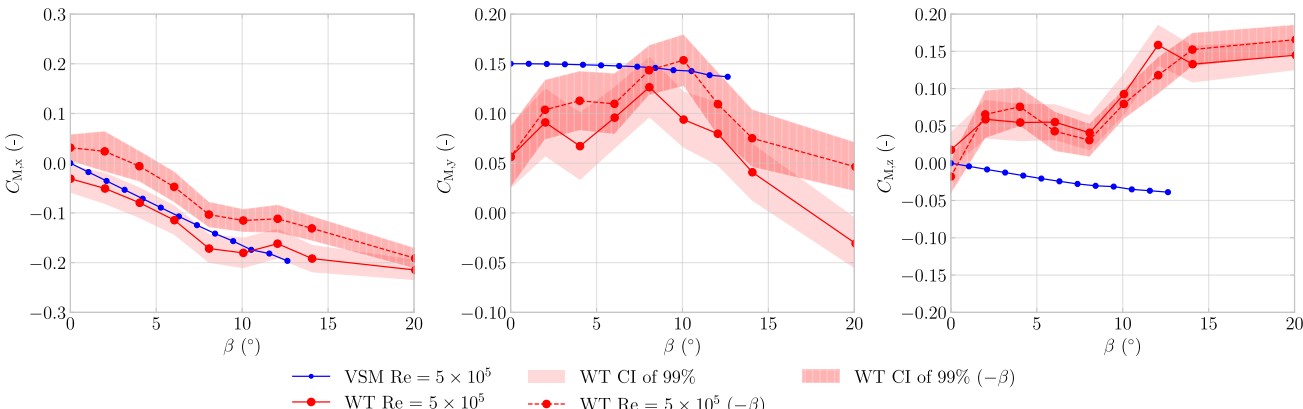

**Figure 13.** Measured moment coefficients together with coefficients computed with VSM simulations, plotted against $\beta$, for $\alpha = 7.4°$.

## 5 Conclusions

This paper presents a wind tunnel investigation of a leading-edge inflatable (LEI) kite, designed as a benchmark case to validate numerical models for airborne wind energy applications. To avoid scaling issues caused by aero-structural deformation, a 1:6.5 rigid scale model of the TU Delft V3 kite was used. The same idealized geometry with fillets, as used in numerical studies, was employed to aid validation. The experiments were conducted in the Open-Jet Facility at TU Delft, with wind tunnel corrections applied to primarily account for downwash effects.

Consistent with findings from previously published 2D simulations, the experimental measurements demonstrate an improvement in aerodynamic performance, i.e. an increase in lift and a reduction in drag, as the Reynolds number increases from $1.3 \times 10^5$ to $5 \times 10^5$.

A zigzag tape was applied to replicate the aerodynamic effect of the stitching seam that connects the canopy to the leading-edge tube. Its height was selected based on theoretical criteria to induce boundary layer transition. At a Reynolds number of $5 \times 10^5$, the addition of the zigzag tape led to a reduction in lift and an increase in drag, aligning with trends reported in the literature. Despite the limited data and kites typically operating at higher Reynolds numbers, the findings suggest that the suction side stitching seam negatively affects the aerodynamic performance.

In the nominal operating regime, the experimental data confirm the force predictions made by the VSM simulations conducted in this study, as well as by previously published RANS CFD simulations. Between 0 and 10° angle of attack, the lift-to-drag ratio $L/D$ remains nearly constant, between 8 and 9. This behavior deviates from conventional wing aerodynamics and warrants careful consideration in kite simulations, as current numerical models are unable to capture the nearly constant trend, likely due to an underestimation of drag in the relevant flow regime.

Within the nominal sideslip range, from $-10$ to $10°$, the experimental results confirm the numerical side force predictions. Given that sideslip conditions inherently arise during turning maneuvers, and that side force plays a critical role in initiating



and sustaining such motions, the observed agreement suggests there is aerodynamic potential within the presented numerical models for accurately predicting steering behavior.

The measurements and simulations differ substantially outside the nominal operating ranges in angle of attack and sideslip. This discrepancy is partly attributed to differences between the wind tunnel conditions and the simulated environment, but also due to the decreasing accuracy of numerical predictions beyond the onset of stall. While the discrepancies indicate potential areas for model refinement, they are not inherently detrimental to accurately predicting kite aerodynamic loads, as they primarily occur outside the nominal operating envelope.

The reported measured values will differ from those of a real kite, as an idealized shape was analyzed and structural deformations were neglected. The actual kite geometry, lacking edge fillets and including a bridle line system, will likely exhibit higher drag. Furthermore, structural deformations such as canopy billowing will alter the aerodynamic performance.

Future work should investigate the causes of the measured asymmetry and aim to reduce uncertainty in moment measurements. To study transition and the influence of the stitching seam in more detail, more refined measurement techniques, e.g. infrared thermography, are recommended. For improved numerical validation, CFD simulations should be conducted at all measured Reynolds numbers and inflow angles, including moment predictions. A particle image velocimetry study was already conducted to analyze the flow fields and enhance understanding. The manuscript is under production and will be published as a companion paper.

*Code and data availability.* The wind tunnel measurements are available Zenodo from https://doi.org/10.5281/zenodo.14288467. The code for the analysis of this data and the generation of the tables and diagrams in this paper is available on Zenodo from https://doi.org/10.5281/zenodo.14930182 and GitHub from https://doi.org/10.5281/zenodo.15316684. This code also includes the VSM simulations performed in the context of this study. The latest version of the VSM can be found on: https://github.com/ocayon/Vortex-Step-Method. The geometric mesh of the TU Delft V3 kite is available on Zenodo from https://doi.org/10.5281/zenodo.15316036 and GitHub from https://github.com/awegroup/TUDELFT_V3_LEI_KITE. More information on the TU Delft V3 Kite is available from https://awegroup.github.io/TUDELFT_V3_KITE/.

*Author contributions.* JAWP compiled the original manuscript, co-designed the experiment, executed the experiment, and performed the analysis. JMvS co-designed the experiment, executed the experiment, performed an initial analysis, and aided in developing figures. Both MG and RS supervised the project, reviewed the manuscript, and contributed to all sections.

*Competing interests.* At least one of the (co-)authors is a member of the editorial board of Wind Energy Science.

*Acknowledgements.* The authors would like to thank the following people for their help: Erik Fritz and David Bensason assisted with the setup of the experimental data processing, and together with René Poland also assisted in the acquisition of the data. Delphine de Tavernier,



for her advice on the planning of the experiment and the feedback on the manuscript. Frits Donker Duyvis, Peter Duyndam, and Dennis Bruikman together resolved all of the technical issues, e.g. cabling repairs. Fabien Schmutz enabled and executed the FARO laser tracker measurement. We would also like to thank Curveworks B.V. for providing a substantial discount and building an excellent scale model.

*Financial support.* This research has been supported by the Nederlandse Organisatie voor Wetenschappelijk Onderzoek (NWO) under grant number 17628. This work has partially been supported by the MERIDIONAL project, which receives funding from the European Union's Horizon Europe Programme under the grant agreement No. 101084216. We acknowledge the use of OpenAI's ChatGPT and Grammarly for assistance in refining the writing style of this manuscript.

## Appendix A: Wind tunnel corrections

### A1    Wind tunnel blockage

Two different effects contribute to the blockage of the flow in the wind tunnel, both affecting the dynamic pressure. There is solid blockage due to the frontal area of the wing and wake blockage arising from momentum loss in the wake downstream of the model. One can estimate the total blockage using the blockage factor, defined as the ratio between the model's frontal area and the jet exit's cross-sectional area (Mercker et al., 1997). With the kite set at the maximum tested angle of attack of $24°$, the projected frontal area $S_\mathrm{f}$ is approximately $0.2\ \mathrm{m}^2$. The octagonal wind tunnel opening has an area $S_\mathrm{n} = 7.47\ \mathrm{m}^2$, resulting in a blockage factor of 3%. For blockage factors below 10%, the open-jet wind tunnel correction model of Lock (1929) has been validated against CFD simulations (Collin, 2019), which states,

$$\frac{\Delta U}{U_\infty} = \tau \lambda \left( \frac{S_\mathrm{f}}{S_\mathrm{n}} \right)^{\frac{3}{2}}, \tag{A1}$$

where $\tau$ represents the tunnel shape factor of approximately 0.22, and $\lambda$ the model shape factor of approximately 0.7, both calculated using the length-to-thickness ratio $c_\mathrm{ref}$ and $h$. The resulting velocity correction is approximately 0.25%.

Barlow et al. (1999) presents another approximation form of the total blockage,

$$\epsilon_\mathrm{t} \approx \frac{S_\mathrm{f}}{4 S_\mathrm{n}}, \tag{A2}$$

with which one finds a correction of 0.67%.

As both methods result in values below 1%, the blockage effects are considered negligible. This aligns with the guidelines of Wickern (2014), which recommend keeping blockage factors below 5%, and Barlow et al. (1999), which advise a maximum of 7.5%.



## A2 Streamline curvature and downwash

The correction model described by Barlow et al. (1999) was used. Although not explicitly stated, it was likely developed for conventional planar wings. The swept-back, highly curved anhedral kite wing is non-planar. In the absence of open-jet tunnel corrections that take dihedral effects into account, the model was assumed valid.

Barlow et al. (1999) defines the total angle correction as the sum of a downwash correction $\Delta\alpha$ and a streamline curvature correction $\Delta\alpha_{\text{sc}}$ in $\mathrm{rad}$,

$$\Delta\alpha_{\text{t}} = \Delta\alpha + \Delta\alpha_{\text{sc}}. \tag{A3}$$

### A2.1 Downwash

The downwash angle correction $\Delta\alpha$ in $\mathrm{rad}$ is calculated using,

$$\Delta\alpha = \delta\frac{A}{C}C_{\text{L}}, \tag{A4}$$

where $A = 0.462$ m$^2$ represents the model reference area by which the model lift coefficient, $C_L$, is defined. The octagonal tunnel jet-exhaust crossectional area is $C = 7.47$ m$^2$. The variable $\delta$ represents an empirically determined factor, given by Barlow et al. (1999) as a function of the wind tunnel geometry and the effective vortex span $b_{\text{e}}$. A $b_{\text{e}} \approx 0.79$ was found using,

$$b_{\text{e}} = \frac{b}{2}\left(1 + \frac{b_{\text{v}}}{b}\right), \tag{A5}$$

where the ratio of the vortex span $b_{\text{v}}$ to geometric span $b = 1.287$ m was found, from Fig. 10.11 on p. 382 in Barlow et al. (1999) using the taper ratio $\lambda_{\text{t}} \approx 0.53$ and the aspect ratio of $\approx 3.5$.

Assuming a near-elliptical loading, the $\delta$ for an octagonal jet can be approximated using the empirical relations of open circular-arc wind tunnel (Rosenhead, 1933; Batchelor, 1944; Gent, 1944). With a ratio of minor to major jet axes $\lambda = 1$, and the ratio of effective span to jet height $k \approx 0.4$, a $\delta \approx -0.126$ was determined from Fig. 10.126 on p. 393 in Barlow et al. (1999).

### A2.2 Streamline curvature

The streamline curvature angle correction $\Delta\alpha_{\text{sc}}$ in $\mathrm{rad}$ is related to the downwash angle correction,

$$\Delta\alpha_{\text{sc}} = \tau_2\Delta\alpha \tag{A6}$$

where $\tau_2$ is an empirically determined factor dependent on whether the wind tunnel has an open or closed test section and the ratio between tail length $l_t$ and tunnel width $2R = 2.85$ m. Barlow et al. (1999) state that, for wings without a defined tail length, one can use a quarter of the chord length instead of the tail length, resulting in $l_t \approx 0.10$ m. With a ratio of 0.035, one finds from Barlow et al. (1999, Fig. 10.37 on p. 400) a $\tau_2 \approx 0.054$. Because the streamline curvature angle correction has a magnitude of roughly 5.4% of the downwash angle correction, it is clear that the downwash correction dominates.





## A3 Total correction

Rewriting the equations and converting from rad to deg, the total angle and load corrections become

$$\Delta\alpha_\mathrm{t} = (1+\tau_2)\delta\frac{A}{C}C_\mathrm{L}\frac{180}{\pi}, \tag{A7}$$

$$\Delta C_\mathrm{D} = \delta\frac{A}{C}C_\mathrm{L}^2, \tag{A8}$$

$$\Delta C_\mathrm{L} = -\Delta\alpha_\mathrm{sc}\frac{dC_\mathrm{L}}{d\alpha}, \tag{A9}$$

$$\Delta C_\mathrm{M,y} = +0.125\Delta\alpha_\mathrm{sc(2)}\frac{dC_\mathrm{L}}{d\alpha}, \tag{A10}$$

where $\Delta\alpha_\mathrm{sc(2)}$ denotes the streamline curvature correction computed using an $l_t$ equal to half the chord length $\tau_{2(2)} \approx 0.108$.
A value of $\frac{dC_\mathrm{L}}{d\alpha} \approx 0.1$ was derived from the experimental results.

Barlow et al. (1999) do not mention any application of their corrections towards the sideways $y$- direction. As the kite, under nonzero sideslip conditions, does produce a non-negligible side force, i.e. roughly 15% of the maximum lift, a downwash and curvature effect might be present. To quantify the effects, it is assumed that Barlow et al. (1999)'s method also holds for the sideways direction in the following form,

$$\Delta\beta_\mathrm{t} = (1+\tau_\mathrm{2,s})\delta\frac{A}{C}C_\mathrm{S}\frac{180}{\pi}, \tag{A11}$$

$$\Delta C_\mathrm{D} = \delta\frac{A}{C}C_\mathrm{S}^2, \tag{A12}$$

$$\Delta C_\mathrm{S} = -\Delta\beta_\mathrm{sc}\frac{dC_\mathrm{S}}{d\beta}, \tag{A13}$$

$$\Delta C_\mathrm{M,z} = +0.125\Delta\beta_\mathrm{sc(2)}\frac{dC_\mathrm{S}}{d\beta}, \tag{A14}$$

with $\tau_\mathrm{2,s} \approx 0.028$ and $\tau_{2(2),s} \approx 0.056$, calculated using the tip chord $c_\mathrm{t} = 0.212$ m. Because $C_\mathrm{S}$ is non-dimensionalized by the
475 same area $A$, and to enable calculations, it is assumed that the same $\delta \approx -0.126$ can be used. A value of $\frac{dC_\mathrm{S}}{d\beta} \approx 0.01$ was derived from the experimental results.

The resulting corrections, similar to the blockage corrections, are deemed negligible when they induce less than 1% change at their maximum—corresponding, for example, to a $0.1°$ shift at a $10°$ angle. Under this criterion, the corrections $\Delta C_\mathrm{L}$, $\Delta C_\mathrm{M,y}$, $\Delta C_\mathrm{S}$, and $\Delta C_\mathrm{M,z}$ are all negligible, each producing a maximum change of less than 0.1%.

An exception is the $\Delta\beta_\mathrm{t}$ correction, which, while still below 1%, approaches the threshold. For instance, in the case shown on the bottom row of Fig. 11, with $C_\mathrm{S} \approx 0.15$, the computed correction is $\Delta\beta_\mathrm{t} \approx 0.068°$, representing approximately 0.97%.

The resulting corrections, similar to the blockage corrections, are deemed negligible if they induce less than 1% change at their maximum, e.g., a $0.1°$ change at $10°$ angle. This renders the $\Delta C_\mathrm{L}$, $\Delta C_\mathrm{M,y}$, $\Delta C_\mathrm{S}$ and $\Delta C_\mathrm{M,z}$ corrections negligible. An exception is $\Delta\beta_\mathrm{t}$, which does not cause 1% change but comes close, e.g. taking the case on the bottom row of Fig. 11 one finds
for a $C_\mathrm{S} \approx 0.15$, a correction of $\Delta\beta_\mathrm{t} \approx 0.068°$, which given $\beta \approx 7°$ implies an 0.97% change.



**Table A1.** Corrections for angle, force, and moment coefficients.

| $\Delta\alpha_t\ (°)$ | $\Delta\beta_t\ (°)$ | $\Delta C_D\ (-)$ |
|---|---|---|
| $-0.47\,C_L$ | $-0.46\,C_S$ | $-0.0078\,C_L^2 - 0.0078\,C_S^2$ |

## Appendix B: Assessment of sensor drift

To ensure consistent data from the load balance measurement device, described in Section 2.3 and shown in Fig. 2, sensor drift was evaluated through repeated measurements. Specifically, 30 s time interval measurements were taken each morning and evening over three consecutive days, corresponding to 12 h intervals. This procedure served to determine whether the drift was substantial enough to influence the results. The measurement drift over time is plotted in Fig. B1, with corresponding mean and standard deviation values reported in Table B1. On average, the standard deviation across the six components (three translational and three rotational) was approximately 1 N. During the experiment, a baseline measurement at near-zero wind

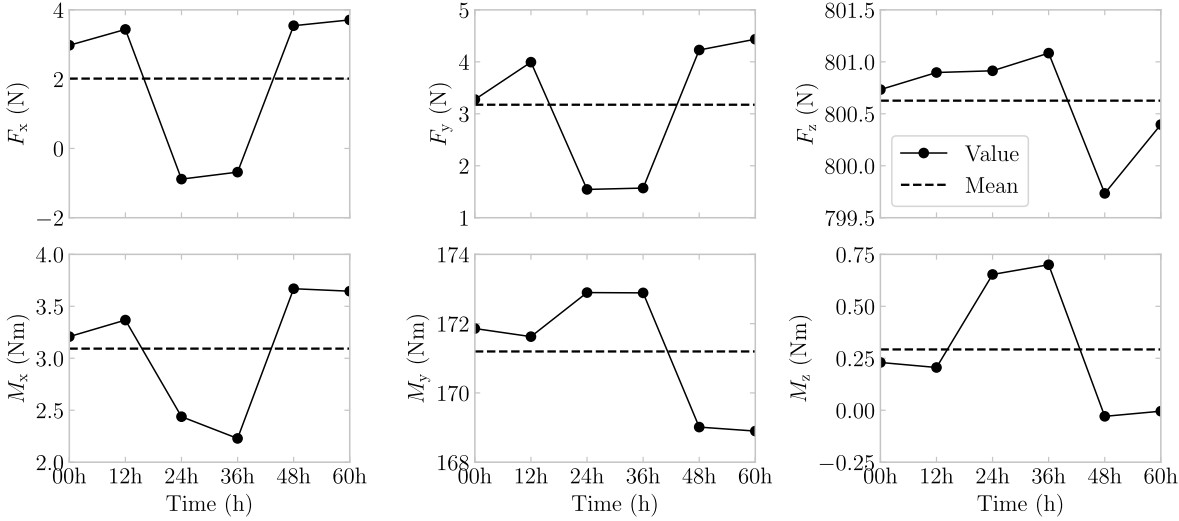

**Figure B1.** Sensor drift of the load balance for the three force components and three moments during the 60 h measurement time.

speed was taken after each change in $\alpha$, followed immediately by a measurement at non-zero $U_\infty$. The aerodynamic load was then obtained by subtracting the baseline from the flow-on measurement. Consequently, sensor drift only affects the resulting data if drift magnitudes occurring over the short interval between the two measurements are comparable to those observed over the 12 h drift assessment intervals.





**Table B1.** Sensor drift mean and standard deviation $\sigma$ values

| Symbol | Unit | Mean | $\sigma$ |
|--------|------|------|----------|
| $F_x$ | N | 2.02 | 1.99 |
| $F_y$ | N | 3.17 | 1.20 |
| $F_z$ | N | 800.63 | 0.45 |
| $M_x$ | $\mathrm{Nm^{-1}}$ | 3.09 | 0.56 |
| $M_y$ | $\mathrm{Nm^{-1}}$ | 171.20 | 1.65 |
| $M_z$ | $\mathrm{Nm^{-1}}$ | 0.29 | 0.29 |

## Appendix C: Support structure loads

To illustrate the relative contribution of the kite and support structure to the total measured loads, the proportions of the measured kite loads and support-structure loads are shown in Fig. C1 for a representative case at $\mathrm{Re} = 5 \times 10^5$ over a $\beta$ sweep; see App. C. Defining the kite load as the signal and the support-structure load as the noise, this ratio serves as a proxy for the signal-to-noise ratio (SNR) and, thus, for measurement uncertainty.

The kite contribution dominates for $C_L$, indicating a high SNR and low associated uncertainty. In contrast, for $C_D$, $C_{M,y}$, and $C_{M,z}$, the support-structure contributions are more significant, implying a lower SNR and correspondingly higher uncertainty.

For proportions in other cases, the reader is referred to the open-source code and open-access dataset, which allow the reproduction of these plots.

## Appendix D: Experimental setup vibration analysis

During the measurements, vibrations were observed and analyzed both qualitatively from video footage and quantitatively using force and moment data sampled at 2000 Hz; see Fig. D1. At $\mathrm{Re} = 6.1 \times 10^5$, the vibrations were deemed potentially destructive under high $\alpha$ and high $\beta$; therefore, some of the intended experiments were not completed. The increasing vibration amplitudes suggest that a natural frequency of the structure or one of its sub-structures was excited, indicating resonance.

As an example, a 1 s data segment at $U_\infty = 25\ \mathrm{ms^{-1}}$, $\alpha = 14°$, and $\beta = 0°$ is shown in Fig. D1, where the force data exhibit high-frequency oscillations, most notably in $F_z$, and the moment data display a resonant trend.

To investigate the resonance behavior observed during testing, the time series data were transformed into the frequency domain using a Fast Fourier Transform (FFT), and the Power Spectral Density (PSD) was computed using a periodogram function. The resulting PSD values were normalized to the range [0, 1] to enable comparison across different wind speeds. For each wind speed, frequency and normalized PSD values were computed for all six channels: three force components ($F_x$, $F_y$, $F_z$) and three-moment components ($M_x$, $M_y$, $M_z$).

To examine the influence of wind speed on the frequency content and to identify potential resonance behavior, the normalized PSDs were plotted up to 100 Hz; see Fig. D2. This frequency range was chosen as the PSD values beyond 100 Hz are negligible



WIND
ENERGY
SCIENCE
DISCUSSIONS
european academy of wind energy
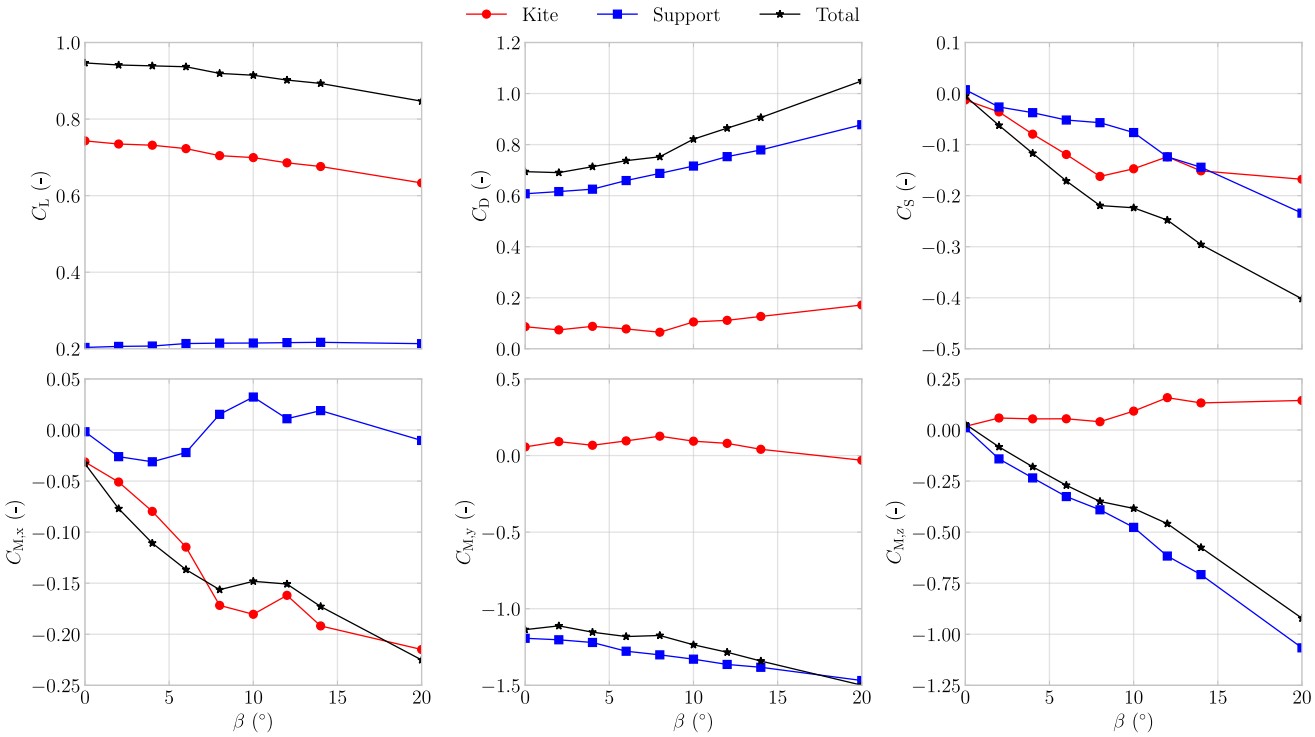

**Figure C1.** Total, support-structure, and kite measured loads plotted for $\mathrm{Re} = 5 \times 10^5$ over a positive $\beta$ sweep, for $\alpha = 7.4°$.

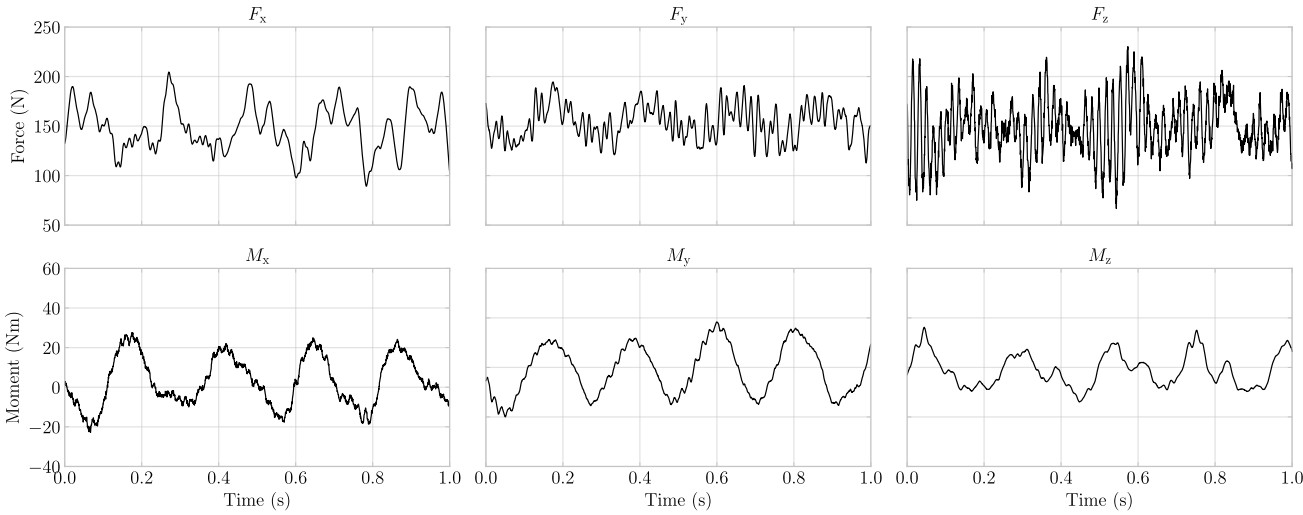

**Figure D1.** Raw measured values at 2000 Hz by the load balance, over a 1 s period taken at 25 ms$^{-1}$ with $\alpha = 15°$ and $\beta = 0°$.




in all channels except $F_z$. As most PSD peaks are concentrated at lower frequencies, the data were also plotted up to 10 Hz;
see Fig. D3.

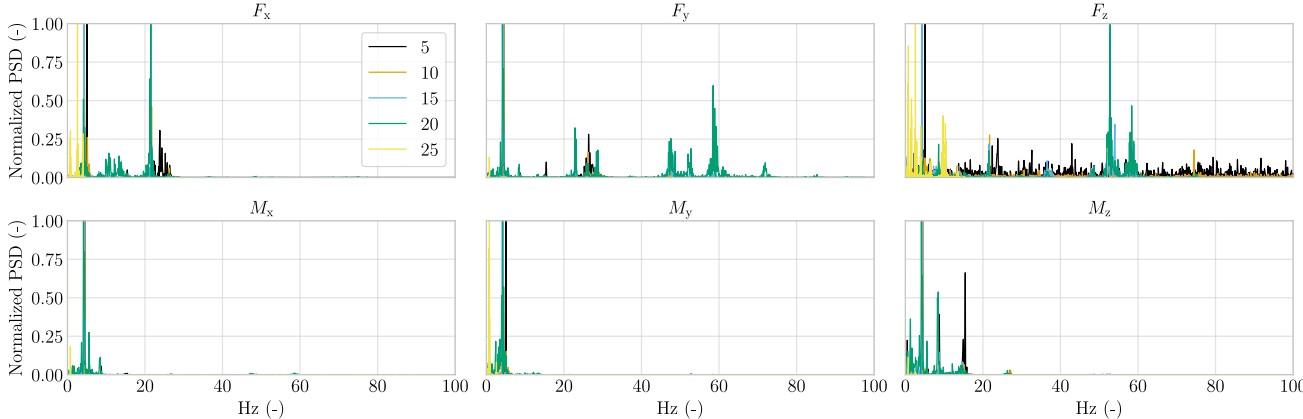

**Figure D2.** Raw measurements transformed into PSD using FFT and a periodogram function and displayed for the three force and moment components up to 100 Hz.

At $U_\infty = 25\,\mathrm{ms}^{-1}$, the number and magnitude of PSD peaks increased, indicating the presence of multiple vibrational modes and aligning with qualitative observations of stronger vibrations. Across most components and flow conditions, a dominant peak was consistently observed at 4–5 Hz, corresponding to the natural frequency of the supporting blue table onto which

the setup was mounted; see Fig. 2 (LeBlanc and Ferreira, 2018). The alignment of these peaks with the structural resonance frequency confirms the occurrence of resonance and explains the elevated uncertainties observed at high $\alpha$ and $\beta$. To avoid introducing filtering-related artifacts and to remain conservative on the uncertainty, it was decided not to filter out the 4–5 Hz band.



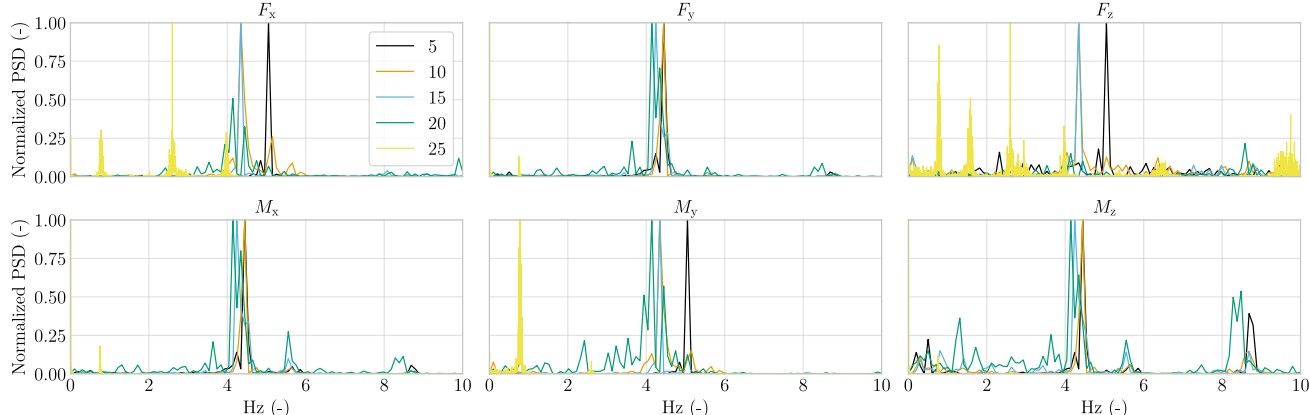

**Figure D3.** Raw measurements transformed into PSD using FFT and a periodogram function and displayed for the three force and moment components up to 10 Hz.

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
