# Peer review of "Wind Tunnel Load Measurements of a Leading-Edge Inflatable Kite Rigid Scale Model"

_Wind Energy Science, 2025_

## Referee Comment (RC2)

**Review of the manuscript wes-2025-77, "Wind Tunnel Load Measurements of a Leading-Edge Inflatable Kite Rigid Scale Model" by J.A.W. Poland, J.M. Spronsen, M. Gaunaa, and R. Schmehl.**

The study presents the wind tunnel measurements of the aerodynamic loads of a downscaled rigid model of a 25 square meter TU Delft V3 Leading-Edge Inflatable (LEI) kite. The experimental measurements are further compared with the results from numerical methods: particularly RANS results from the literature and VSM method. Authors claim to have observed similar trends and values in the load data between the experimental data and the results from numerical methods under nominal operational conditions. Confidence intervals are presented for the measured data, and the discrepancies are attributed to the experimental setup in the wind tunnel. Correction of the experimental measurements are performed using standard methods considering blockage, downwash, and streamline curvature. The measured data and used codes are made publicly available through Zenodo.

Introduction along with the literature review is presented in the first part of the study, which is then followed by the description of the experimental setup and downscaled model. Mean and standard deviation of the measured aerodynamic loads as a function of angle of attack and side slip angle, considering uncertainty analysis, boundary layer transition and Reynolds number effects are presented.

The manuscript is well written and organized. The study provides novel experimental data sets on steady aerodynamic coefficients for the downscaled models of LEI kite. However, the authors do not discuss/acknowledge about the unsteady loads or aerodynamic coefficients, which are responsible for the realistic aerodynamic performance of LEI kite. In addition, the following comments should be addressed for improving the quality of manuscript.

1. Page 8, Line 134: A measurement period of 10 s is used. Is the time sufficient for the convergence of statistics? Discussions on the convergence analysis should be made.
2. Page 8, Figure 6: Please check the x-label of the figure, $Re \times 10^{-5}$ or $Re \times 10^{5}$?
3. Page 9, Line 155: where $U_k$ is the local velocity at the roughness height, which may be approximated by $U_\infty$. Does the presence of boundary layer near the leading edge affects the approximation of $U_k$ as $U_\infty$?
4. Page 10, Line 184: $x_{cg}$ is repeated twice and needs to be corrected with $z_{cg}$. Does the change of angle of attack, slide slip angle as well as deformation of the model at high test speed affects these values? If it does, how are they corrected?
5. Page 12, Table 3: What is the angle of attack and slide slip angle for these measurements?
6. Page 14, Line 269: Please discuss why $\alpha = 7.4°$ was selected.
7. Page 16, Line 293: '.. we applied the area ratio of 3.7 as a correction factor to the Cs reported in Vire et al. What's the reason for using this specific ratio? It should be clarified.

8. Page 16, Line 301: '.. contained a 1.02 degree offset in the angle of attack". How has this been decided? Is it done for the better match of the experimental results with the results from numerical simulation?
9. Page 17, Line 324: In figure 10, the legend does not indicate with and without strut CFD studies.
10. Page 17, Line 329: 'All numerical models predict a higher maximum L/D…' Is this because of the difference in Reynolds number between the experimental setups and numerical simulations?
11. Page 21, Line 377, 'the observed agreement suggests there is an aerodynamic potential within the presented numerical models..'. However, the experimental results show that there is a disagreement after a certain range of angle of attack and slide slip angle. The statement should be justified with details.
12. Page 28, Line 523: Comments on the sources of vibrations (resonance, vortex shedding) will make the manuscript more robust.

---

## Author Comment (AC2)

[revised manuscript text omitted]

 examined in detail. Measured aerodynamic forces and moments  were compared with numerical simulations
to assess the consistency between experimental and computational results.

The remainder of this paper is organized as follows. Section 2 describes the experimental methodology. Section 3 presents the results of our wind tunnel tests, focusing on analyzing the uncertainties and the effect of Reynolds number. A discussion on the agreement with numerical predictions follows in Sect. 4, and the conclusions are presented in Sect. 5 along with recommendations for future work.

**2    Experimental methodology**

This section first discusses the specifics of the wind tunnel and the scale model. This is followed by a description of the experimental setup, the measurement matrix, zig-zag tape measurements, and the data processing method, including the required wind tunnel corrections.

**2.1    Open Jet Facility**

The wind tunnel experiments were conducted in the Open Jet Facility (OJF) at the Faculty of Aerospace Engineering of Delft University of Technology from 1 to 10 April 2024. The facility is a closed-loop wind tunnel, featuring an octagonal jet exhaust nozzle with maximum dimensions of $2.85 \times 2.85$ m, and a contraction ratio of 3:1, as illustrated in Fig 2. The jet discharges into a test section room with dimensions 13 m in width and 8 m in height. The wind tunnel is equipped with a 500 kW electric motor driving a large fan, which generates a controlled streamwise velocity of up to 35 ms$^{-1}$ in the test section. Corner vanes and wire meshes guide the flow to ensure uniform flow conditions, resulting in a turbulence intensity of 0.5% in the test section (Lignarolo et al., 2014).

**2.2    Rigid scale model**

As the original TU Delft LEI V3 kite is 8.3 m wide and the width of the OJF exhaust nozzle is only 2.85 m, a scale model had to be used. With the main purpose of the measurement campaign being the acquisition of validation data for numerical tools, the scale model was manufactured to match the wing geometry used in earlier CFD simulations (Viré et al., 2022). This geometry  was adapted from the original  design CAD model to facilitate mesh smoothness in the simulations. Notably, the bridle line system was omitted, the trailing edge connecting the upper and lower canopy surfaces  was rounded, and an edge fillet  was applied at all  canopy–tube junctions. The only difference between the CFD and manufactured geometries is the use of a canopy with increased thickness for structural integrity—3 to 4 mm instead of 1 mm. The model geometry was verified using a laser tracker with a spatial resolution of  0.5 µm (FARO, 2024). Figure 3 compares the manufactured physical model with the  rendered geometry and the overlaid laser-tracked outline of the physical model. The agreement between the manufactured and rendered geometry was

Manually adjustable horizontal bar

Support structure

Load balance

Rotary table

within 1 mm in chord, height, and width, corresponding to errors of less than 0.25% in all cases, as detailed in Table 1.

[Figure]

**Figure 3.** Rigid scale model of the TU Delft LEI V3 kite: (a)  photograph of the model, rotated by 180°, with its back facing the blue octagonal OJF exhaust nozzle; (b)  rendering of the model  from a similar perspective, with  the laser-tracked outline overlaid in red, and the reference chord $c_{\text{ref}}$, height $h$, and width $w$ indicated in white.

[revised manuscript text omitted]
  lies within the boundary layer and is therefore different, i.e. generally somewhat smaller, than the external velocity $U_\infty$; nonetheless, it is often approximated by $U_\infty$  for practical purposes (Driest and McCauley, 1960; Tani, 1969). For a more precise assessment, the local velocity profile within the boundary layer could be employed to determine $U_\mathrm{k}$ directly; however, this typically necessitates either supplementary measurements or detailed boundary-layer computations, which were not available in the present study. The resulting functional dependency of $k$ on the Reynolds number defined by Eq. (1) is shown in Fig. 6 for two different values of $\mathrm{Re_{k,crit}}$. The diagram also includes the selected tape height of 0.2 mm to trigger transition from approximately $\mathrm{Re} \geq 3.9 \times 10^5$ according to the estimate $\mathrm{Re_{k,crit}} = 200$. The tape, produced by Glasfaser Flugzeug-Service GmbH with a 60° tooth angle, was applied at 5%

[Figure]

**Figure 6.** Required minimal trip height versus Re for different values of $\mathrm{Re_{k,crit}}$.

chord, following the approach in Soltani et al. (2011); Gahraz et al. (2018); Dollinger et al. (2019); De Tavernier (2021).

**2.6 Data post-processing**

The measured load data were converted to the non-dimensional aerodynamic coefficients as follows:

205  1. subtract zero-wind measurements,

2. non-dimensionalize the load data,

3. translate the coordinate system, from the load balance origin $O$ to the center of gravity of the scale model

4. correct for sideslip,

5. subtract non-dimensionalized support structure loads,

210  6. apply wind tunnel corrections.

(1) First, the zero-wind measurements taken before every $\alpha$ change were subtracted to eliminate background noise from the signals, including the structure's weight and sensor drift.

(2) In the next step, the measurements were  non-dimensionalised using the air density $\rho$,  determined at each measurement point, varying from 1.14 to 1.19 $\mathrm{kg\,m^{-3}}$; the inflow

215  speed $U_\infty$ ; the projected area $A$; and the reference chord $c_{\mathrm{ref}}$ of the scale model, as listed in Table 1. The forces $F_{\mathrm{i}}$ and moments $M_{\mathrm{i,b}}$  were non-dimensionalized using

$$C_{\mathrm{i}} = \frac{2F_{\mathrm{i}}}{\rho U_\infty^2 A}, \qquad\qquad i = 1,2,3,$$

$$C_{\mathrm{M,i,b}} = \frac{2M_{\mathrm{i,b}}}{\rho U_\infty^2 A c_{\mathrm{ref}}}, \qquad\qquad i = 1,2,3.$$

(3) To represent the moment coefficients in the wing reference frame, they had to be translated from the load-balance measurement center to the center of gravity CG of the scale model.  When the mid-span chord-line is aligned with the $x$-axis, the CG is located at $-0.172$ m in $x$- and $-0.229$ m in $z$-direction with respect to the mid-span trailing-edge point, see Fig. 4. The distance from the origin $O$ to the CG varied with angle of attack but remained constant with sideslip, as the load balance was mounted atop the rotary table and therefore rotated with it. The rolling moment coefficient $C_{M,x,b}$ is translated using,

$$C_{M,x} = C_{M,x,b} - C_{F,y} z_{cg}. \tag{3}$$

The pitching- and yawing-moment coefficients, $C_{M,y}$ and $C_{M,z}$, respectively,  were determined as

$$C_{M,y} = -C_{M,y,b} + C_{F,z} x_{cg} - C_{F,x} z_{cg}, \tag{4}$$

$$C_{M,z} = -C_{F,z,b} - C_{F,y} x_{cg}. \tag{5}$$

In these expressions, $x_{cg}$, $y_{cg}$ and  $z_{cg}$ are the coordinates of the scale model's center of gravity, with respect to $O$.

(4) Because the  load balance was mounted on top of the rotary table, and $y$ is defined perpendicular to the incoming flow, the force and measured moment coefficients had to be corrected for the sideslip. The force and moment coefficient vectors  were transformed, at each sideslip angle $\beta_i$, through matrix multiplication with the rotation matrix $\mathbf{R}$:

$$\mathbf{R} = \begin{bmatrix} \cos\beta & \sin\beta & 0 \\ -\sin\beta & \cos\beta & 0 \\ 0 & 0 & 1 \end{bmatrix}. \tag{6}$$

(5) To isolate the aerodynamic forces of the kite, measurements were made with only the support structure. These measurements were performed at the minimum, mean, and maximum $\alpha$ values. Missing data points were determined by interpolation, which was carried out by fitting two linear segments from the minimum to the mean and from the mean to the maximum, respectively.

The aerodynamic loads on the support structure only were measured and processed through steps (1) to (4) such that the resulting aerodynamic coefficients could then be subtracted from the coefficients of the kite including the support structure. It was critical to non-dimensionalize before subtracting these measurements, as atmospheric conditions could not be assumed constant throughout the experiment. Specifically, during the experiment, the temperature varied between 20 and 32°C.

(6) The last step entailed applying the wind tunnel corrections that arise from blockage, streamline curvature, and downwash or upwash in both $y$- and $z$-directions. For a detailed analysis of these effects, the reader is referred to App. B. The conclusions  were that with a blockage factor of 3%, the corrections due to blockage are negligible, which aligns with the recommendations of Wickern (2014) to keep the blockage factor below 5% and of Barlow et al.

(1999) to stay below 7.5%. Following Barlow et al., the corrections due to streamline curvature and downwash were

250 calculated and found to be non-negligible, shown in Table B1, and hence applied.

**3 Results**

This section first addresses measurement uncertainties, followed by the effect of forced boundary layer transition. Subsequently, the aerodynamic force and moment coefficients are presented as functions of the angle of attack and sideslip angle, with the Reynolds number as parameter.

255 ### 3.1 Uncertainty analysis

This section quantifies the main sources of measurement uncertainty to ensure data reliability and repeatability, including sensor drift, support-to-kite load proportion, vibration analysis, coefficient of variation, and measurement repeatability. Although a load balance sensor drift was detected, it was concluded not to affect the results, as detailed in App. C. Analyzing the proportions of support-structure loads to kite loads as signal-to-noise ratio, one finds high

260 certainty for lift and lower for $C_{M,y}$, $C_{M,z}$, as detailed in App. D.

For some measurements at $U_\infty = 25~\mathrm{ms}^{-1}$ and high values of $\alpha$ and $\beta$, the wind tunnel model started to vibrate considerably. To avoid physical damage, these specific measurements were not completed, which is why some data points are missing at $\mathrm{Re} = 6.1 \times 10^5$. A vibration analysis revealed structural resonance at 4–5 Hz, close to the resonance frequency of the supporting blue table (see Fig. 2) as reported in LeBlanc and Ferreira (2018). This

265 frequency band was not filtered to avoid introducing processing artifacts. See App. E for further details, i.e., time series and power spectral density analyses.

The coefficient of variation, denoted as CV, offers a dimensionless metric for comparing variability across different datasets by normalising the standard deviation relative to the mean (Pearson, 1896). For each aerodynamic force or moment coefficient,

270
$$\mathrm{CV}_{i} = \frac{\overline{\sigma_i}}{\mu_i}, \quad i \in \{\mathrm{L}, \mathrm{D}, \mathrm{S}, \mathrm{M_x}, \mathrm{M_y}, \mathrm{M_z}\}$$

where $\overline{\sigma_i}$ is the average standard deviation of coefficient $i$, and $\mu_i$ is its mean value, both computed over the ensemble of measurements. Table 3 lists $\mathrm{CV}_i$ for each Re, except for $\mathrm{Re} = 6.1 \times 10^5$, which is excluded due to incomplete data. The means $\mu_i$ were computed over the full range of $\alpha$ and $\beta$ for $\mathrm{CV_L}$, $\mathrm{CV_D}$, and $\mathrm{CV_{M,y}}$. For $\mathrm{CV_S}$, $\mathrm{CV_{M,x}}$, and $\mathrm{CV_{M,z}}$, only positive values of $\beta$ were considered to avoid

275 including near-zero loads at $\beta = 0°$, which could lead to inflated values of $\mathrm{CV}_i$ and skew the statistical averages.

~~The decline in $\mathrm{CV}_i$ values from $\mathrm{Re} = 1.3$ to $2.5 \times 10^5$ reflects a reduction in measurement uncertainty, likely due to an improved signal-to-noise ratio at higher wind speeds. Force measurements generally exhibit lower relative uncertainty compared to moment measurements, which is attributed to their inherently lower signal-to-noise ratios. The cases at $\mathrm{Re} = 3.8$ and $5 \times 10^5$ show the smallest values of $\mathrm{CV}_i$, indicating the highest measurement precision.~~

**Table 3.** Coefficient of variation $CV_i$ of the data for varying Re.

| $Re \times 10^5$ $(-)$ | 1.3 | 2.5 | 3.8 | 5 |
|---|---|---|---|---|
| $CV_L$ | 1.11 | 0.35 | 0.17 | 0.15 |
| $CV_D$ | 0.84 | 0.54 | 0.53 | 0.58 |
| $CV_S$ | 1.27 | 0.94 | 0.89 | 0.90 |
| $CV_{M,x}$ | 5.67 | 2.28 | 2.18 | 2.31 |
| $CV_{M,y}$ | 33.40 | 8.43 | 4.54 | 5.33 |
| $CV_{M,z}$ | 2.90 | 2.54 | 2.90 | 2.24 |

280    The decline in $CV_i$ values from $Re = 1.3$ to $2.5 \times 10^5$ reflects a reduction in relative measurement uncertainty, as the standard deviation becomes smaller relative to the mean. In this work, force measurements exhibited lower relative uncertainty compared to moment measurements, which can be attributed to their inherently higher signal-to-noise ratios. The cases at $Re = 3.8$ and $5 \times 10^5$ exhibit the smallest values of $CV_i$, indicating the highest relative measurement precision. However, it should be noted that a low CV reflects only the precision—that is, the

285    spread or random uncertainty of the measurements—and does not account for possible systematic errors, or constant offsets, that may affect accuracy.

At $Re = 5 \times 10^5$, $\alpha = 5.7°$ and $\beta = -20, 0$ and $20°$ measurements were made three times to check the repeatability. For each of these measurements, the standard deviation within these repeated measurements $\sigma_{rm}$ is shown in Table 4. The authors conclude that the measurement repeatability is overall high by comparing the orders of magnitude of

290    the averaged standard deviation, $1 \times 10^{-1}$, and the repeatability standard deviation, $1 \times 10^{-4}$. Smaller uncertainties show for $\beta = 0°$, affecting the lift coefficient $C_L$ and the pitching moment $C_{M,y}$ the most.

**Table 4.** Standard deviations of the repeatability measurements $\sigma_{rm}$ for three $\beta$ values taken with $\alpha = 5.7°$.

| | $\sigma_{rm} \times 10^{-4}$ | | |
|---|---|---|---|
| | $\beta = -20°$ | $\beta = 0°$ | $\beta = 20°$ |
| $C_L$ | 2.793 | 0.699 | 2.562 |
| $C_D$ | 0.076 | 0.085 | 0.014 |
| $C_S$ | 0.085 | 0.030 | 0.300 |
| $C_{M,x}$ | 1.903 | 1.030 | 2.585 |
| $C_{M,y}$ | 7.034 | 1.899 | 6.254 |
| $C_{M,z}$ | 0.222 | 0.120 | 0.766 |

**3.2 Effect of forced boundary layer transition**

The measured aerodynamic force coefficients with and without zigzag tape are plotted in Fig. 7 for $\beta = 0°$ and
$\alpha = 9.4°$, excluding the $\text{Re} = 6.1 \times 10^5$ case due to missing data. In addition to the mean values, a confidence interval
295 (CI) is plotted, indicating with 99% certainty that the mean lies within the given range. As detailed in Sec. 2.4, the
load balance records data over a 10 s time interval, thereby capturing between 125 and 625 fluid parcels passing
through. The resulting samples are regarded as temporally correlated; one supporting argument is that each fluid
element traverses the measurement region over 16 to 80 ms, while data are sampled at much finer 0.5 ms intervals.
To accurately estimate the sample measurement uncertainty of this correlated time series, the heteroskedasticity
300 and autocorrelation-consistent (HAC) estimator by Newey and West (1987) is employed. The method requires an
estimate of the time lag. A time lag of 11 samples was found from taking the integer value of $N_{\text{samples}}^{\frac{1}{4}}$ (Greene,
2019).

[Figure]

**Figure 7.** Aerodynamic force coefficients plotted with standard deviation, with and without zigzag tape, at $\text{Re} = 1.3$, 2.5,
3.8 and $5 \times 10^5$, at an averaged corrected $\alpha = 9.4°$.

 For context, the theoretical analysis leading to Fig. 6  indicated that a zigzag tape height
of  0.2 mm would be insufficient to force transition at $\text{Re} = 2.5 \times 10^5$,
305  marginally sufficient at $3.8 \times 10^5$, and  sufficient at $5 \times 10^5$. The three horizontally separated regions
in Fig. 7 correspond to these different Re, with black and red symbols indicating measurements obtained without
and with zigzag tape, respectively. At $\beta = 0°$, adding zigzag tape resulted in higher lift and lower drag
for $\text{Re} = 2.5$ and $3.8 \times 10^5$,  whereas the opposite trend  was observed at $\text{Re} = 5 \times 10^5$, including a
 12% increase in drag. This  latter observation is consistent with  the literature,
310 where the introduction of zigzag tape led to decreased lift and increased drag (Gahraz et al., 2018; Zhang et al.,
2017a, b; Dollinger et al., 2019).

Definitive conclusions cannot be drawn for the non-zero sideslip cases, where data is limited to $\text{Re} = 3.8 \times 10^5$ at $\alpha = 9.4°$ for $\beta = \pm 10°$. Nonetheless, based on the measured increase in lift and side force, along with an observed 50% reduction in drag, the authors hypothesize that, in the sideslip configuration, the zigzag tape may locally promote a laminar-to-turbulent transition that delays flow separation.

Without zigzag tape and  under sideslip, the measured $C_S$ value  remains near zero, independent of Re. In contrast, with zigzag tape, a negative $C_S$ is observed at $\text{Re} = 5 \times 10^5$.  This difference suggests that the zigzag tape introduces a setup asymmetry, possibly due to imperfect tape application.

**3.3 Reynolds number effects**

Figure 8  presents the measured force and moment coefficients as functions of $\alpha$ for  various Re.  In the measurements, the lift coefficient $C_L$ increases with Re,  but so does the drag coefficient $C_D$, resulting in a $C_L/C_D$ ratio that does not show a consistent increasing trend with Re, except for the $\text{Re} = 6.1 \times 10^5$ case. This finding contrasts with the 3D numerical simulations of Viré et al. (2022), in which both an increase in lift and a decrease in drag were observed from $\text{Re} = 1 \times 10^5$ to $1 \times 10^6$, leading to higher $C_L/C_D$ with increasing Re. The absence of a similar trend in the present measurements may be attributed to the smaller range of Re realised experimentally, differences in canopy thickness, and/or drag underprediction in the simulations, as discussed in Sect. 4.

The  $\text{Re} = 1.3 \times 10^5$ case exhibits the  least smooth curves, attributable to a less favourable signal-to-noise ratio, i.e.,  the load magnitudes are relatively small compared to the  support-structure loads. The $C_L$–$\alpha$ plot suggests that, compared to higher Re,  stall development may occur at lower angles, as evidenced by the earlier decrease in slope of the $C_L$–$\alpha$ curve for $\text{Re} = 1.3 \times 10^5$, indicating reduced lift growth, and hence the possible onset of local flow separation. This aligns with aerodynamic theory predicting earlier separation in laminar flows due to lower sensitivity to adverse pressure gradients (Anderson, 2016).

 Although small in magnitude, the non-zero values of $C_S$  $C_{M,x}$ and $C_{M,z}$ may have indicated an asymmetry in the setup. For coefficients of smaller magnitude, the non-smoothness of the curves was amplified, which was consistent with the higher relative uncertainties found, as

345 indicated by the coefficient of variation in Table 3. The pitching moment coefficient $C_{\mathrm{M,y}}$   exhibited an increasing trend with increasing $\alpha$.

[Figure]

**Figure 8.** Aerodynamic force and moment coefficients plotted against $\alpha$ for Re varying from $1.3 \times 10^5$ to $6.1 \times 10^5$ at $\beta = 0°$.

 The influence of $\beta$,  shown in Fig. 9  , was examined at $\alpha = 7.4°$, as this condition most closely resembled the
350 average angle of attack of $8°$ observed during the reel-out phase of a V3 kite flight (Cayon et al., 2025). A perfectly symmetric setup  would yield coefficients $C_{\mathrm{L}}$, $C_{\mathrm{D}}$, and  $C_{\mathrm{M,y}}$ symmetric about $\beta = 0°$,  and coefficients $C_{\mathrm{S}}$,  $C_{\mathrm{M,x}}$, and $C_{\mathrm{M,z}}$ antisymmetric. In practice, slight asymmetries in the   experimental setup caused small deviations from this ideal symmetry, notably non-zero values of $C_{\mathrm{S}}$,  $C_{\mathrm{M,x}}$, and
355 $C_{\mathrm{M,z}}$ at $\beta = 0°$, as well as minor asymmetries  of $C_{\mathrm{L}}$, $C_{\mathrm{D}}$, and  $C_{\mathrm{M,y}}$ about the vertical axis. The largest

deviations from ideal symmetry were observed at the lowest $\text{Re} = 1.3 \times 10^5$, with overall increasing symmetry as Re increased.

 The $C_\text{L}$ plot reveals an overall  increase in lift coefficient with increasing Re, consistent with the findings when varying $\alpha$. Notably, around $\beta = 8°$, the $C_\text{S}$ curve exhibits both positive and negative peaks, suggesting a non-linear relationship with $\beta$. At this same angle, a local maximum is observed in $C_\text{M,x}$ for the $\text{Re} = 5 \times 10^5$ case, and off-trend behavior shows for $C_\text{D}$ and $C_\text{M,z}$. Similar off-trend behavior near $\beta = \pm 8°$ also appears in $\beta$ sweeps at other values of $\alpha$. The potential underlying causes of this phenomenon are examined further in Sect. 4.

[revised manuscript text omitted]

Although this study provides a rigorous evaluation and benchmarking of numerical models through direct comparison with carefully acquired experimental data, the simulations are not yet considered fully validated. Strict validation

would require comprehensive assessment across multiple geometries, operating conditions, and Reynolds numbers, as well as resolution of the identified limitations to ensure reliable predictive capability across the full operational envelope.

The reported measured values will differ from those of a real kite, as an idealised shape was analysed. The actual kite geometry, lacking edge fillets and  incorporating a bridle line system, will likely exhibit higher drag. Furthermore, structural deformations such as canopy billowing  and unsteady aerodynamic loads will further alter the aerodynamic response.

Future work should investigate the causes of the measured asymmetry and aim to reduce uncertainty in moment measurements. To study transition and the influence of the stitching seam in more detail, more refined measurement techniques, e.g. infrared thermography, are recommended. For improved numerical validation, CFD simulations should be conducted at all measured Reynolds numbers and inflow angles, including moment predictions. A particle image velocimetry study was already conducted to analyze the flow fields and enhance understanding. The manuscript is under production and will be published as a companion paper.

*Code and data availability.* The geometric mesh of the TU Delft V3 kite is available on Zenodo from https://doi.org/10.5281/zenodo.15316036 and through https://awegroup.github.io/TUDELFT_V3_KITE/docs/datasets.html. The wind tunnel measurements are available on Zenodo from https://doi.org/10.5281/zenodo.14288467. The code for the analysis of this data and the generation of the tables and diagrams in this paper is available on Zenodo from https://doi.org/10.5281/zenodo.14930182 and GitHub from Zenodo https://doi.org/10.5281/zenodo.15316684 or directly through GitHub https://github.com/jellepoland/WES_load_wind_tunnel_measurements_TUDELFT_V3_LEI_KITE. This code uses the latest version of the Vortex Step Method (VSM) to perform simulations, available on GitHub: https://github.com/ocayon/Vortex-Step-Method.

This paper includes verified computational reproducibility, confirmed through an independent CODECHECK process, which is an open science initiative to improve reproducibility (Nüst and Eglen, 2021). The certificate is accessible through: https://doi.org/10.5281/zenodo.15603144.

*Author contributions.* JAWP compiled the original manuscript, co-designed the experiment, executed the experiment, and performed the analysis. JMvS co-designed the experiment, executed the experiment, performed an initial analysis, and aided in developing figures. Both MG and RS supervised the project, reviewed the manuscript, and contributed to all sections.

*Competing interests.* At least one of the (co-)authors is a member of the editorial board of Wind Energy Science.

*Acknowledgements.* The authors would like to thank the following people for their help: Erik Fritz and David Bensason assisted with the setup of the experimental data processing, and together with René Poland also assisted in the acquisition of the data. Delphine de Tavernier, for her advice on the planning of the experiment and the feedback on the manuscript. Frits Donker Duyvis, Peter Duyndam, and Dennis Bruikman together resolved all of the technical issues, e.g. cabling repairs. Fabien Schmutz enabled and executed the FARO laser tracker measurement. We would also like to thank Curveworks B.V. for providing a substantial discount and building an excellent scale model.

*Financial support.* This research has been supported by the Nederlandse Organisatie voor Wetenschappelijk Onderzoek (NWO) under grant number 17628. This work has partially been supported by the MERIDIONAL project, which receives funding from the European Union's Horizon Europe Programme under the grant agreement No. 101084216. We acknowledge the use of OpenAI's ChatGPT and Grammarly for assistance in refining the writing style of this manuscript.

**Appendix A: Statistical convergence of measurement period**

A measurement duration of 10 s was selected based on the characteristic aerodynamic time scale of the system, defined as the time required for a fluid element to traverse the kite's reference chord. For each tested condition, this corresponded to approximately 125 to 625 independent flow passages within the 10 s interval, depending on the free-stream velocity. This ensured that statistical averages were derived from a sufficiently large number of uncorrelated samples, thereby mitigating the influence of temporally correlated fluctuations.

To assess statistical convergence, key measurement conditions—namely $\alpha = 5.7°$ at $U = 20$ m.s$^{-1}$ and $\beta = -20°$, 0°, and 20°—were repeated three times. The close agreement in both mean and fluctuating load coefficients across these repetitions confirmed that a 10 s sampling window was adequate to obtain converged statistics under the present steady-state aerodynamic conditions. While longer sampling durations may be necessary for capturing slower or rare unsteady phenomena, the selected interval was found to be appropriate for the regime investigated.

To further substantiate this, a convergence analysis was performed using both running average and block analysis techniques. As shown in Fig. A1, these methods revealed only marginal fluctuations in the computed statistics over the 10 s window, thereby validating the statistical robustness of the chosen measurement duration.

**Appendix B: Wind tunnel corrections**

**B1 Wind tunnel blockage**

Two different effects contribute to the blockage of the flow in the wind tunnel, both affecting the dynamic pressure. There is solid blockage due to the frontal area of the wing and wake blockage arising from momentum loss in the wake downstream of the model. One can estimate the total blockage using the blockage factor, defined as the ratio between the model's frontal area and the jet exit's cross-sectional area (Mercker et al., 1997). With the kite set at

[Figure]

**Figure A1.** Running average and block average analysis of the 10 s, over a sample showing the forces in the $z$- and $x$-axis. Demonstrating that the selected period is sufficiently long to achieve a statistically converged average.

the maximum tested angle of attack of 24°, the projected frontal area $S_f$ $A_f$ at $\alpha = 24°$ is approximately 0.2 m². The octagonal wind tunnel opening has an area $S_n = 7.47$ m², resulting in a blockage factor of 3%. For blockage factors below 10%, the open-jet wind tunnel correction model of Lock (1929) has been validated against CFD simulations (Collin, 2019), which states,

$$\frac{\Delta U}{U_\infty} = \tau \lambda \left( \frac{S_f}{S_n} \frac{A_f}{S_n} \right)^{\frac{3}{2}}, \tag{B1}$$

where $\tau$ represents the tunnel shape factor of approximately 0.22, and $\lambda$ the model shape factor of approximately 0.7, both calculated using the length-to-thickness ratio $c_{ref}$ and $h$. The resulting velocity correction is approximately 0.25%.

Barlow et al. (1999) presents another approximation form of the total blockage,

$$\epsilon_t \approx \frac{S_f}{4S_n} \frac{A_f}{4S_n}, \tag{B2}$$

with which one finds a correction of 0.67%.

As both methods result in values below 1%, the blockage effects are considered negligible. This aligns with the guidelines of Wickern (2014), which recommend keeping blockage factors below 5%, and Barlow et al. (1999), which advise a maximum of 7.5%.

**B2   Streamline curvature and downwash**

The correction model described by Barlow et al. (1999) was used. Although not explicitly stated, it was likely developed for conventional planar wings. The swept-back, highly curved anhedral kite wing is non-planar. In the absence of open-jet tunnel corrections that take dihedral effects into account, the model was assumed valid.

Barlow et al. (1999) defines the total angle correction as the sum of a downwash correction $\Delta\alpha$ and a streamline curvature correction $\Delta\alpha_{\mathrm{sc}}$ in rad,

$$\Delta\alpha_{\mathrm{t}} = \Delta\alpha + \Delta\alpha_{\mathrm{sc}}. \tag{B3}$$

**B2.1   Downwash**

The downwash angle correction $\Delta\alpha$ in rad is calculated using,

$$\Delta\alpha = \delta\frac{A}{C}C_{\mathrm{L}}, \tag{B4}$$

where $A = 0.462$ m$^2$ represents the model reference area by which the model lift coefficient, $C_L$, is defined. The octagonal tunnel jet-exhaust crossectional area is $C = 7.47$ m$^2$. The variable $\delta$ represents an empirically determined factor, given by Barlow et al. (1999) as a function of the wind tunnel geometry and the effective vortex span $b_{\mathrm{e}}$. A $b_{\mathrm{e}} \approx 0.79$ was found using,

$$b_{\mathrm{e}} = \frac{b}{2}\left(1 + \frac{b_{\mathrm{v}}}{b}\right), \tag{B5}$$

where the ratio of the vortex span $b_{\mathrm{v}}$ to geometric span $b = 1.287$ m was found, from Fig. 10.11 on p. 382 in Barlow et al. (1999) using the taper ratio $\lambda_{\mathrm{t}} \approx 0.53$ and the aspect ratio of $\approx 3.5$.

Assuming a near-elliptical loading, the $\delta$ for an octagonal jet can be approximated using the empirical relations of open circular-arc wind tunnel (Rosenhead, 1933; Batchelor, 1944; Gent, 1944). With a ratio of minor to major jet axes $\lambda = 1$, and the ratio of effective span to jet height $k \approx 0.4$, a $\delta \approx -0.126$ was determined from Fig. 10.126 on p. 393 in Barlow et al. (1999).

**B2.2   Streamline curvature**

The streamline curvature angle correction $\Delta\alpha_{\mathrm{sc}}$ in rad is related to the downwash angle correction,

$$\Delta\alpha_{\mathrm{sc}} = \tau_2\Delta\alpha \tag{B6}$$

[revised manuscript text omitted]

**Table C1.** Sensor drift mean and standard deviation $\sigma$ values

| Symbol | Unit | Mean | $\sigma$ |
|--------|------|------|----------|
| $F_{\mathrm{x}}$ | N | 2.02 | 1.99 |
| $F_{\mathrm{y}}$ | N | 3.17 | 1.20 |
| $F_{\mathrm{z}}$ | N | 800.63 | 0.45 |
| $M_{\mathrm{x}}$ | $\mathrm{Nm}^{-1}$ | 3.09 | 0.56 |
| $M_{\mathrm{y}}$ | $\mathrm{Nm}^{-1}$ | 171.20 | 1.65 |
| $M_{\mathrm{z}}$ | $\mathrm{Nm}^{-1}$ | 0.29 | 0.29 |

**Appendix D: Support structure loads**

To illustrate the relative contribution of the kite and support structure to the total measured loads, the proportions of the measured kite loads and support-structure loads are shown in Fig. D1 for a representative case at $\mathrm{Re} = 5 \times 10^5$ over a $\beta$ sweep; see App. D. Defining the kite load as the signal and the support-structure load as the noise, this ratio serves as a proxy for the signal-to-noise ratio (SNR) and, thus, for measurement uncertainty.

The kite contribution dominates for $C_{\mathrm{L}}$, indicating a high SNR and low associated uncertainty. In contrast, for $C_{\mathrm{D}}$, $C_{\mathrm{M,y}}$, and $C_{\mathrm{M,z}}$, the support-structure contributions are more significant, implying a lower SNR and correspondingly higher uncertainty.

For proportions in other cases, the reader is referred to the open-source code and open-access dataset, which allow the reproduction of these plots.

**Appendix E: Experimental setup vibration analysis**

During the measurements, vibrations were observed and analyzed both qualitatively from video footage and quantitatively using force and moment data sampled at 2000 Hz; see Fig. E1. At $\mathrm{Re} = 6.1 \times 10^5$, the vibrations were deemed potentially destructive under high $\alpha$ and high $\beta$; therefore, some of the intended experiments were not completed. The increasing vibration amplitudes suggest that a natural frequency of the structure or one of its sub-structures was excited, indicating resonance.

As an example, a 1 s data segment at $U_\infty = 25~\mathrm{ms}^{-1}$, $\alpha = 14°$, and $\beta = 0°$ is shown in Fig. E1, where the force data exhibit high-frequency oscillations, most notably in $F_{\mathrm{z}}$, and the moment data display a resonant trend.

To investigate the resonance behavior observed during testing, the time series data were transformed into the frequency domain using a Fast Fourier Transform (FFT), and the Power Spectral Density (PSD) was computed using a periodogram function. The resulting PSD values were normalized to the range [0, 1] to enable comparison across different wind speeds. For each wind speed, frequency and normalized PSD values were computed for all six channels: three force components ($F_{\mathrm{x}}$, $F_{\mathrm{y}}$, $F_{\mathrm{z}}$) and three-moment components ($M_{\mathrm{x}}$, $M_{\mathrm{y}}$, $M_{\mathrm{z}}$).

[Figure]

**Figure D1.** Total, support-structure, and kite measured loads plotted for $\mathrm{Re} = 5 \times 10^5$ over a positive $\beta$ sweep, for $\alpha = 7.4°$.

[Figure]

**Figure E1.** Raw measured values at 2000 Hz by the load balance, over a 1 s period taken at 25 ms$^{-1}$ with $\alpha = 15°$ and $\beta = 0°$.

To examine the influence of wind speed on the frequency content and to identify potential resonance behavior, the normalized PSDs were plotted up to 100 Hz; see Fig. E2. This frequency range was chosen as the PSD values beyond 100 Hz are negligible in all channels except $F_z$. As most PSD peaks are concentrated at lower frequencies, the data were also plotted up to 10 Hz; see Fig. E3.

[Figure]

**Figure E2.** Raw measurements transformed into PSD using FFT and a periodogram function and displayed for the three force and moment components up to 100 Hz.

At $U_\infty = 25$ ms$^{-1}$, the number and magnitude of PSD peaks increased, indicating the presence of multiple vibrational modes and aligning with qualitative observations of stronger vibrations. Potential sources of the observed vibrations include structural resonance, wherein the natural frequencies of the experimental setup are excited by unsteady aerodynamic loads, and vortex shedding from the model or its mounting components, which can introduce periodic forcing. Both mechanisms are known to amplify dynamic responses in wind tunnel experiments, particularly at elevated angles of attack and higher wind speeds. Across most components and flow conditions, a dominant peak was consistently observed at 4–5 Hz, corresponding to the natural frequency of the supporting blue table onto which the setup was mounted; see Fig. 2 (LeBlanc and Ferreira, 2018). The alignment of these peaks with the structural resonance frequency confirms the occurrence of resonance and explains the elevated uncertainties observed at high $\alpha$ and $\beta$. To avoid introducing filtering-related artifacts and to remain conservative on the uncertainty, it was decided not to filter out the 4–5 Hz band.

**Appendix F:  Angle of attack offset correction**

An offset of 1.02 ° in the angle of attack was identified in the original V3 kite CAD geometry. This offset originated from a geometric inconsistency: the vector from the mid-span leading edge to trailing edge was tilted upward by

[Figure]

**Figure E3.** Raw measurements transformed into PSD using FFT and a periodogram function and displayed for the three force and moment components up to 10 Hz.

1.02 ° relative to the intended horizontal reference plane, as shown in Fig. F1. All subsequent simulations in this work were corrected by applying this offset to maintain alignment with the conventional aerodynamic reference frame.

[Figure]

**Figure F1.** Geometric verification of the angle-of-attack offset in the original CAD geometry. Each of the three images stacked vertically illustrates the LEI airfoil of the V3 kite at mid-span. Black lines indicate cross-sectional slices through the leading edge, while red lines represent slices through the trailing edge. The visible vertical mismatch between these lines confirms the presence of the offset.

685

This misalignment was inadvertently propagated into earlier RANS-CFD studies, including those by Viré et al. (2020) and Viré et al. (2022). As a result, the angles of attack reported in those publications do not strictly adhere to the standard aerodynamic definition, namely the angle between the incoming flow and the chord line.

The issue was discovered during the present wind tunnel campaign. In subsequent discussions with G. Lebesque—whose MSc thesis formed the basis of (Viré et al., 2022)—it was confirmed that the offset had gone unnoticed at the time. This was further substantiated through cross-sectional geometric inspection, where lines connecting the leading and trailing edges showed a clear tilt relative to a horizontal reference, see Fig. F1.

To resolve this discrepancy, the geometry has been corrected to eliminate the offset. Updated and verified CAD files have been made publicly available at: https://awegroup.github.io/TUDELFT_V3_KITE/docs/datasets.html.

[revised manuscript text omitted]